# Unpacking the paradoxical impact of ethical leadership on employees' pro-social rule-breaking behavior: The interplay of employees' psychological capital and moral identity

Mushtaq Ahmed[ID]*[◯], Muhammad Ishfaq Khan[ID][◯]

Management Sciences Department, Capital University of Science & Technology, Islamabad, Pakistan

◯ These authors contributed equally to this work.
* mushtaq1635@hotmail.com

**Data Availability Statement:** All relevant data are within the paper and its Supporting Information files.

## Abstract

This study delves into the complex dynamics of ethical leadership's influence on employees' pro-social rule-breaking behavior, taking into account the mediating role of psychological capital and the moderating effect of moral identity. Using data collected from nursing staff in Pakistani hospitals and analyzed through PLS SEM, the study yielded unexpected results. Contrary to the initial hypotheses, the findings reveal a positive relationship between ethical leadership and employees' pro-social rule-breaking behavior within organizational settings. Furthermore, the study identifies psychological capital as a key mediator in this relationship, while moral identity emerges as a crucial moderator. These results challenge the conventional perception of ethical leadership as an exclusively positive form of leadership and underscore its unintended consequences. Moreover, they underscore the significance of employees' psychological processes and individual differences in unraveling this paradoxical relationship. These results have the potential to reshape how organizations view ethical leadership and consider the unintended outcomes it may generate. Future research can build upon these findings to explore the boundaries and contextual factors that influence the effects of ethical leadership, ultimately contributing to a more comprehensive understanding of leadership dynamics in diverse organizational settings.

## Introduction

Ethical leadership (EL) significantly influences employees' behavior in organizations by fostering a culture of integrity and ethical conduct. Defined as "demonstration of normatively appropriate conduct through personal actions and interpersonal relationships, and the promotion of such conduct to followers through two-way communication, reinforcement, and decision-making" [1, p.120], EL builds trust and encourages employees to adopt similar values. Ethical leaders prioritize employee well-being and fairness, leading to higher job satisfaction

**Funding:** The author(s) received no specific funding for this work.

**Competing interests:** The authors confirm that this research received no financial support, and they have no competing interests to disclose.

and engagement [2, 3]. They make resilient, adaptable decisions that consider the welfare of all stakeholders and embrace corporate social responsibility [4, 5]. This approach deters unethical behavior, safeguarding the organization's legal and reputational standing, and contributes to its long-term success by fostering ethical practices and responsible decision-making [6, 7].

Pro-social rule-breaking (PSRB) is a type of constructive deviance where employees intentionally violate organizational policies to benefit organizational stakeholders [8, 9]. PSRB has been defined as "any instance where an employee intentionally violates a formal organizational policy with the primary intention of promoting the welfare of the organization or one of its stakeholders" [10, p. 6]. PSRB fosters innovation, creativity, and initiative, which are crucial for organizational growth and competitiveness. However, it also poses risks, such as ethical or legal violations that can damage the organization's reputation, undermine employee cooperation, and create compliance issues [13, 14].

Identifying key predictors of employee behavior within organizational settings has been a primary focus of management research [15]. The existing literature highlights the significance of leadership style in forecasting constructive deviance behaviors in organizations [11, 16]. Furthermore, most leadership studies predominantly concentrate on Western contexts, resulting in substantial gaps in understanding leadership dynamics in non-Western cultural settings [17]. Despite the critical importance of PSRB, there is a notable scarcity of comprehensive studies investigating the specific role of EL in shaping these behaviors. Ethical leaders, who exhibit normatively appropriate conduct and promote ethical behavior through their actions and decision-making, have the potential to influence employees' tendencies to engage in PSRB [15]. By understanding the nuanced ways in which EL impacts PSRB, organizations can more effectively balance the encouragement of innovative, beneficial rule-breaking with the maintenance of ethical and legal standards.

Moreover, contemporary studies have recognized that employees' behavior is influenced by affective, cognitive, and behavioral processes [18, 19]. Within this context, psychological capital (PsyCap) has gained significance as an essential explanatory mechanism that encompasses these facets in leadership and employee behavior. PsyCap is defined as: "an individual's positive psychological state of development and is characterized by: (1) having confidence (self-efficacy) to take on and put in the necessary effort to succeed at challenging tasks; (2) making a positive attribution (optimism) about succeeding now and in the future; (3) persevering toward goals and, when necessary, redirecting paths to goals (hope) in order to succeed; and (4) when beset by problems and adversity, sustaining and bouncing back and even beyond (resilience) to attain success" [20]. However, despite the recognition of PsyCap as a pivotal psychological resource, the precise function of PsyCap as a mediating mechanism within the link between EL and employees' PSRB remains unexplored. Examining PsyCap as a mediator can offer insights into how EL influences the psychological and cognitive processes of employees, thereby affecting their engagement in PSRB.

Furthermore, employees' moral identity (MID) is acknowledged as a crucial moderator influencing the relationship amongst leadership, employees' psychological processes and behaviors [21–23]. MID can be defined as "an individual's self-conception organized around a set of moral traits" [24]. Notwithstanding the recognition of MID as a crucial individual trait, its potential role as a moderator in the link between EL and PsyCap has not been examined. Investigating MID as a moderator helps to understand the boundary conditions of EL's effectiveness, providing a more comprehensive and contextually relevant understanding of how leadership influences employees behavior within organizations.

Finally, while social cognitive theory (SCT) [25] traditionally emphasizes cognitive and social cognitive processes, contemporary workplaces present multifaceted challenges that may extend beyond SCT's original scope. Therefore, exploring SCT's applicability in management

and organizational behavior contexts can provide valuable insights into how cognitive processes interact with diverse organizational structures, leadership styles, and social environments to shape employee behavior and organizational outcomes. Therefore while SCT traditionally centers on cognitive and social cognitive processes, its application in understanding real-world workplace behaviors requires further examination within the realms of management and organizational behavior [26].

The study aims to address these research gaps through four key objectives. First, it investigates the impact of EL on employees' PSRB, particularly in non-Western cultural contexts, expanding our knowledge of EL's influence in diverse organizations. Second, it delves into the explanatory mechanism of employees' PsyCap in the EL-PSRB relationship, offering insights into how EL encourages PSRB. Third, it explores when employees' MID moderates the connection between EL and PsyCap, highlighting the conditions where EL's impact on PsyCap is most pronounced. Finally, it evaluates the applicability of SCT in understanding contemporary workplace behaviors, contributing to a more profound understanding of employee conduct. These objectives collectively fill research gaps and provide valuable insights for scholars and practitioners in management and organizational behavior.

This study holds significant implications on multiple fronts. First, it challenges the traditional view of EL as exclusively positive form of leadership by uncovering its unexpected impacts on employees' PSRB, enriching our comprehension of EL's influence. Second, it expands our knowledge of EL's influence on employees' PSRB in a non-Western context, highlighting a positive connection and broadening the application of leadership theories in non-Western cultural settings. Third, it emphasizes the crucial role of employees' PsyCap as a mediator in the EL-PSRB relationship, underlining the importance of psychological processes in EL contexts. Fourth, it reveals when employees' MID moderates the link between EL and PsyCap, offering insights into individual differences that shape workplace behavior in leadership research. Lastly, the application of SCT provides a fresh perspective on the intricate connection between EL and PSRB, emphasizing the interplay of cognitive processes, self-regulation, and contextual factors in the realm of leadership and ethics.

## Theory and hypotheses development

This framework, rooted in SCT [25], highlights the significant role of external influences, including leadership, in shaping individuals' cognitive and psychological processes. EL, characterized by its commitment to ethical values, profoundly influences employees' perceptions, attitudes, and beliefs about ethical behavior, which can subsequently influence their behavior, including PSRB [4, 5].

SCT emphasizes the dynamic, bidirectional relationship between individuals and their environment. Employees not only passively receive influence but actively contribute to their surroundings. In the context of EL, employees' behaviors, including their engagement in PSRB, can influence the leadership they encounter. If employees perceive that their ethical leader values and supports their PSRB, this perception may reinforce and encourage such behavior [27–29].

This comprehensive framework considers the interplay of environmental influences, psychological processes, and the demonstrated behavior. It integrates EL as the external environmental component, PsyCap representing cognitive and psychological processes, individual differences encapsulating MID, and workplace PSRB characterizing employees' behavior within the organization [25, 30].

Grounding our theoretical framework in SCT provides a robust basis for examining the relationships and dynamics investigated in our study [31]. The theoretical model is illustrated in Fig 1.

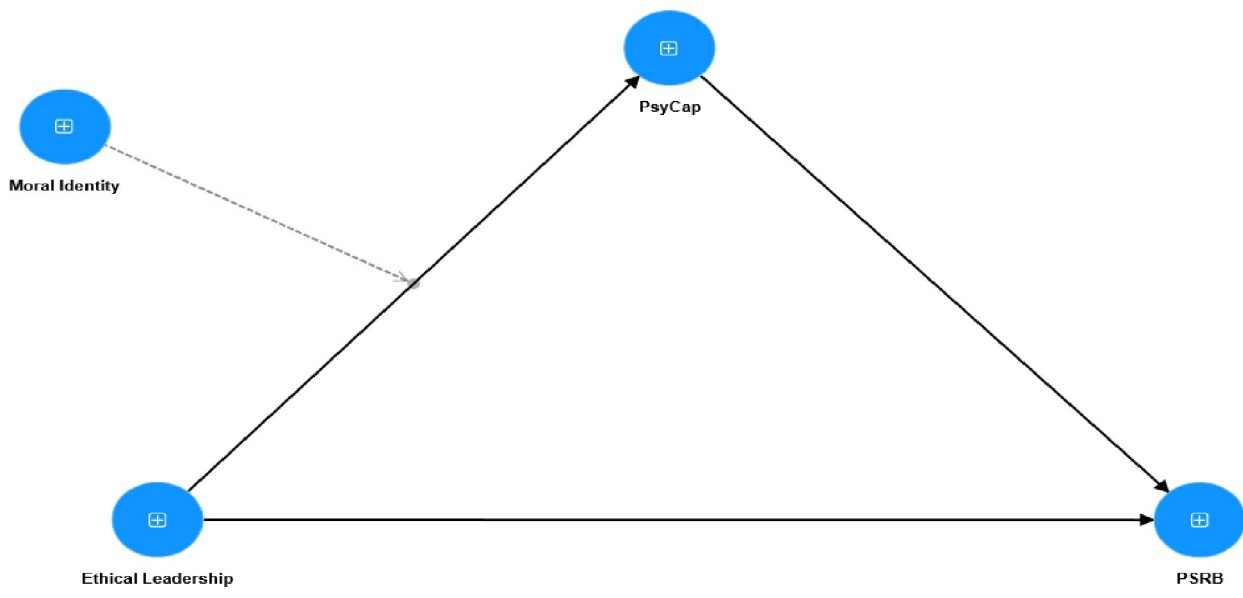

**Fig 1. Theoretical model.**

### Ethical Leadership (EL) and employees' Pro-Social Rule Breaking (PSRB)

EL is crucial in organizational management, involving leaders who exemplify and promote appropriate behavior among employees [1]. EL fosters a positive work environment built on trust and fairness, motivating employees to engage in pro-social behaviors, which boosts job satisfaction and overall organizational success [4, 6]. Employees who view their leaders as ethical feel more valued and committed, leading to higher morale and productivity [2, 32]. However, the impact of EL on constructive deviance behaviors, such as PSRB, is debated. While EL encourages adherence to ethical standards, it may also empower employees to take initiative and deviate from rules if it benefits the organization [5, 33]. This dual influence underscores the complexity of EL in shaping employee behavior, balancing beneficial innovation with maintaining ethical integrity. Understanding this balance is essential for effective organizational management.

PSRB involves employees intentionally breaking organizational rules to benefit the organization or its stakeholders [8, 9]. On one hand, PSRB can drive innovation, creativity, and initiative—key elements for organizational growth and adaptation in competitive markets [10, 34, 35]. These positive outcomes are crucial for fostering a dynamic and responsive organizational culture. Employees who engage in PSRB often do so out of a sense of commitment to the organization, taking proactive steps to address problems or improve processes, which can lead to significant advancements and competitive advantages.

On the contrary, engaging in PSRB can lead to breaches of legal and ethical standards, undermining trust, cooperation, and tarnishing the organization's reputation [8, 13, 36]. Despite being labeled as pro-social [9, 10, 37], concerns arise regarding the paradoxes and unintended consequences linked with such behaviors, including potential rule-breaking, injustice, dishonesty, and ethical norm violations [38–40]. Furthermore, employees involved in PSRB may feel compelled to assist others, potentially conflicting with organizational goals and their own interests [41–43]. Therefore, the challenge for organizations is to balance the innovative and beneficial aspects of PSRB with the need to maintain ethical and legal standards, ensuring that such behaviors do not lead to detrimental outcomes.

Different leadership styles affect PSRB in diverse ways. Research has shown that servant leadership and inclusive leadership positively influence PSRB by fostering supportive, compassionate, and inclusive work environments [44, 45]. These leadership styles encourage employees to feel psychologically safe, valued, and supported, which can promote pro-social behaviors, including PSRB. Conversely, research on paternalistic leadership in China reveals that its moral dimension negatively impacts PSRB, although other dimensions like authoritarian leadership show positive associations [46]. This suggests that while moral guidance is essential, an overly moralistic approach may hinder employees' willingness to engage in constructive deviance. The distinctive negative relationship found between the moral leadership dimension of paternalistic leadership and employees' PSRB introduces a layer of complexity that underscores the need for a nuanced understanding of how various leadership styles impact PSRB.

Zhu and colleagues [12] offer insights into how EL, ethical idealism, and cultural values interact to shape PSRB in East Asia. Their findings challenge the universally positive view of EL, highlighting the influence of cultural nuances. In the specific cultural context of China, EL may not always be perceived positively, suggesting that cultural values significantly influence the impact of EL on employee behavior. Recognizing these cultural sensitivities is crucial for understanding EL's impact on PSRB. This highlights the significance of considering cultural contexts when examining the effects of EL on employees' behaviors.

According to SCT [25], individuals learn by observing and imitating role models, such as leaders. Ethical leaders serve as role models by displaying ethical behavior and actively promoting and endorsing ethical standards. Employees observe their leaders' ethical actions and incorporate these principles into their own behavior. This observational learning process, reinforced by the ethical conduct of leaders, contributes to a reduction in PSRB within the organization [4, 47].

Moreover, ethical leaders enhance employees' self-regulation through effective communication and reinforcement of ethical principles, further diminishing the propensity for rule-breaking [28, 48]. By providing clear ethical guidelines and consistently reinforcing them, ethical leaders help employees internalize these standards, leading to a decrease in behaviors that violate established norms.

EL also acts as a buffer against ethical pressures. Ethical leaders who exhibit ethical behavior offer guidance and support to employees encountering ethical dilemmas, assisting them in navigating complex situations and making decisions that are consistent with organizational values [49]. This supportive role of ethical leaders helps mitigate the pressure employees might feel to engage in PSRB, ensuring that their acts remain aligned with the organization's ethical standards. Based on these theoretical insights and empirical results, we propose the following hypothesis:

**Hypothesis 1:** Ethical leadership is negatively related to the employees' pro-social rule breaking within in the organization.

## Mediating role of employees' Psychological Capital (PsyCap)

PsyCap represents a positive psychological state in individuals, encompassing elements such as hope, self-efficacy, resilience, and optimism [20, 50]. It represents an individual's motivational orientation rooted in positive psychological processes, which are dynamic, developmental, and distinct from stable personality traits [20, 50]. These components synergistically contribute to the higher-order construct of PsyCap, which significantly influences outcomes in organizational settings [50, 51].

PsyCap is recognized as a critical psychological mechanism explaining the link between leadership behaviors and employee outcomes [18, 19, 52]. Empirical studies have consistently shown that PsyCap acts as a mediating mechanism between EL and various organizational

outcomes. For example, Bouckenooghe and colleagues (2015) demonstrated that EL positively impacts job performance through mechanisms involving PsyCap and goal congruence [53]. Similarly, recent research by Goswami and colleagues (2023) found that PsyCap mediates the associations between EL and knowledge sharing, as well as knowledge creation [54]. These findings collectively suggest that fostering PsyCap through EL practices leads to desirable organizational outcomes.

Grounded in SCT [25], EL exercises a profound influence on employees' PsyCap through mechanisms primarily involving observational learning. SCT posits that individuals learn behaviors and attitudes through observation and modeling of others, especially those in leadership roles [25]. Observational learning within the context of EL occurs as employees observe and internalize ethical behaviors demonstrated by their leaders [31, 52]. These observations play a pivotal role in shaping employees' perceptions and beliefs about ethical conduct and organizational values. By witnessing ethical behaviors, employees develop a heightened awareness of ethical norms and principles, which in turn contributes to the development and enhancement of their PsyCap [31, 55].

PsyCap, comprising components such as hope, self-efficacy, resilience, and optimism [28, 30], is nurtured through this observational learning process. As employees perceive and absorb the ethical behaviors of their leaders, they experience an increase in self-efficacy—the belief in their ability to execute tasks and achieve desired outcomes [28, 30]. This enhanced self-efficacy, coupled with improved self-regulation abilities to manage their emotions and behaviors effectively, fosters a positive and elevated psychological state among employees [28, 30].

Moreover, this heightened PsyCap not only empowers employees with a sense of capability and competence but also imbues them with the confidence to positively influence their organizational environment and society at large [56]. Employees who feel psychologically empowered are more likely to undertake actions aligned with organizational goals and values, demonstrating a proactive approach in contributing to the greater good of the organization [54, 55, 57]. Hence, this sense of resilience fosters a readiness to engage in strategic initiatives, such as PSRB, as a means of contributing to the greater good, demonstrating a strategic response aligned with organizational objectives [54, 55, 58]. Therefore, based on the theoretical framework of SCT and empirical evidence, we propose the following hypothesis:

**Hypothesis 2:** Employees' psychological capital positively mediates the relationship between ethical leadership and employees' pro-social rule breaking in the organization.

## Moderating role of employees' Moral Identity (MID)

MID is a fundamental aspect of an individual's character that profoundly influences the link between leadership and employees' psychological processes and behaviors [24]. It serves as a moral compass, guiding an individual's moral reasoning and actions within complex organizational contexts. MID acts as a moral anchor, encouraging alignment of behavior with deeply held ethical principles [59, 60]. It extends beyond mere moral awareness, shaping how employees perceive and respond to ethical challenges, decisions, and dilemmas in their professional lives. Moreover, MID promotes consistency between personal moral values and behavior, contributing to the maintenance of ethical conduct within organizations [61, 62].

Numerous studies have explored how MID moderates the impact of EL on employees' psychological processes and behavior, yielding a variety of findings. For instance, Al Halbusi and colleagues (2023) discovered that EL positively influences employees' moral behavior, especially when individuals possess both high MID and strong self-control [21]. This suggests that employees with a high sense of MID and self-discipline are more inclined to respond positively to ethical leadership by engaging in morally sound behaviors.

Similarly, Wang and colleagues (2021) found that followers with elevated MID and strong leader identification perceive ethical leaders as role models, leading to a reduction in unethical behavior. Conversely, individuals with lower MID and weaker leader identification might display more unethical behavior despite the presence of ethical leadership [15]. This indicates that the effectiveness of EL in promoting ethical behavior is significantly influenced by the moral attributes of the followers.

Contrarily, Chuang and Chiu's study (2018) revealed that moral personality can weaken the link between EL and employee voluntary behaviors, particularly among individuals with high idealism. Their research indicates that the effect of moral personality is influenced by relativism, emphasizing the role of moral ideology in predicting deviant behavior [63]. This highlights how EL's impact on employee behavior is contingent upon individual moral attributes and ideologies.

Further, Moore and colleagues (2019) demonstrated that EL reduces employees' tendency to morally disengage, resulting in fewer unethical decisions and deviant behaviors. They noted that the impact of EL is moderated by employees' MID, leading to different effects [64]. For individuals with a weak MID, EL acts as a "saving grace," helping to curb unethical tendencies. In contrast, for those with a strong MID, EL creates a "virtuous synergy," enhancing the alignment between their moral identity and ethical behavior.

Grounded in SCT [25], individuals are influenced by their cognitive processes, which include personal standards, self-efficacy beliefs, and MID. MID reflects the importance of moral values in an individual's self-concept, shaping how they perceive and react to ethical behaviors. In an EL context, leaders demonstrate ethical behaviors and values, acting as role models for their employees [2, 4]. Employees with an elevated MID are more inclined to resonate with these ethical practices because they align with their own deeply held moral values. This congruence between the leader's ethical actions and the employee's MID enhances the employee's cognitive and emotional engagement with the organization. As a result, these employees experience increased self-efficacy, optimism, hope, and resilience, which are key components of PsyCap [28, 30, 58].

Therefore, for individuals with a strong MID, the relationship between EL and PsyCap is amplified; EL not only influences their behavior but also significantly boosts their psychological resources. Conversely, employees with lower MID may not experience the same level of cognitive and emotional alignment with ethical leaders, resulting in a weaker impact on their PsyCap [21, 23, 63]. This moderating effect of MID highlights the crucial role of employees' moral values in determining the extent to which EL can enhance employees' PsyCap. Thus, fostering a strong MID among employees can amplify the positive effects of EL, leading to more profound psychological and organizational benefits [16, 21]. Therefore, based on the theoretical framework of SCT and empirical evidence, we propose the following hypothesis:

**Hypothesis 3:** Employees' moral identity moderates the relationship between ethical leadership and employees' psychological capital such that the relationship is stronger for employees with higher moral identity than low.

## Methods

### Population and sampling

The primary objective of this study was to investigate the association between EL and PSRB among employees in the healthcare industry of Pakistan. The population of this study comprised the registered nurses employed in both public and private hospitals across Pakistan. Nursing staff are well-suited to assess EL perceptions because they prioritize patient care, sometimes involving rule-breaking for the greater good. Their patient-centered roles, ethical

sensitivity, and professional training make them apt evaluators of EL. To ensure the generalizability of the findings, the research was conducted in public and private hospitals of capital and provincial cities of Islamabad, Rawalpindi, Lahore, Peshawar, Karachi, and Quetta, ensuring diverse perspective within the nation's healthcare industry. We employed purposive sampling technique, a non-probability sampling method, for data collection as sampling frame of all public and private hospitals in those cities was not available. This choice was driven by the practical constraints inherent in accessing such an exhaustive list.

## Sample size

The sample size choice followed statistical guidelines and prior research. Using the GPower formula with an effect size of 0.05 and a power of 0.90, one arrow towards the endogenous construct, the minimum sample size was 150. However, given the complexity of methods like PLS-SEM, Memon and colleagues [65] recommended a sample size range of 160 to 300. However, Comrey and Lee [66] suggested a sample size 50 as very poor, 100 as poor, 200 as fair, 300 as good, 500 as very good, and 1000 as excellent. Similarly, a higher sample size is suggested by the scholars for PLS SEM [67, 68]. Therefore, a sample size of 515 was deemed suitable for the current study. Through higher sample size, the study sought to achieve robust population representation, mitigate potential sampling errors, and augment statistical power for the detection of significant relationships among variables.

## Data collection

The data collection procedures conformed to the recommendations outlined by Aguinis and colleagues [69]. Our engagement with hospital management involved leveraging personal and professional networks, supported by an official university authorization letter for data collection. This approach effectively garnered official approval and also secured support from top-level management. The dissemination and retrieval of questionnaires from nursing staff was facilitated by the heads of administrative departments at each hospital. The respondents were selected randomly from a list presented by the designated point of contact.

Data collection was executed through the use of survey-based questionnaires, ensuring a consistent approach for measuring study constructs and minimizing potential measurement errors. The administration of these surveys was conducted in English, as it is the official language for communication in Pakistan. Moreover, the survey targeted nursing staff members who held at least a bachelor's degree in nursing with minimum one year experience.

A cover letter was appended to the survey instruments, serving the purpose of explaining the research objectives, guaranteeing the confidentiality of participants' responses, and underscoring the voluntary nature of their involvement in the research. Importantly, participants were explicitly informed that it was not necessary to disclose their personal identities or the names of their respective healthcare institutions. Furthermore, a systematic coding system was implemented to enable the alignment of responses across the three temporal phases. Additionally, to facilitate a better comprehension of the survey, clear definitions of each study variable were presented at the outset of each section, aiding participants in selecting the most appropriate response.

We employed a time-lagged cross-sectional research design, comprising three separate data collection phases at eight-week intervals. Each phase focused on different aspects of the research model, aiming to minimize the risk of common method bias [70]. The time-lagged approach enabled us to make inferences regarding the temporal relationships among the variables under study. As part of quality control measures, we meticulously verified the questionnaire codes for consistency across all three data collection waves. The cumulative response rate across these three phases stood at 65.66%, with a valid response rate of 57.22%. These response rates align

**Table 1. Hospital wise response rate.**

| Cities | Hospitals | Questionnaires Distributed | Questionnaires Received | Response Rate (%) |
|---|---|---|---|---|
| Islamabad | Public | 75 | 53 | 70.67 |
| | Private | 75 | 52 | 69.33 |
| Rawalpindi | Public | 75 | 51 | 68 |
| | Private | 75 | 50 | 66.67 |
| Lahore | Public | 75 | 51 | 68 |
| | Private | 75 | 50 | 66.67 |
| Karachi | Public | 75 | 49 | 65.33 |
| | Private | 75 | 50 | 66.67 |
| Peshawar | Public | 75 | 46 | 61.33 |
| | Private | 75 | 48 | 64 |
| Quetta | Public | 75 | 45 | 60 |
| | Private | 75 | 46 | 61.33 |

with established benchmarks for studies involving time-lagged data [71]. The data statistics are reflected as hospital wise response rate in Table 1, total response rate in Table 2, valid response rate in Table 3, sample characteristics in Table 4 and descriptive statistics in Table 5 below.

## Measurements

The research framework comprises four key constructs to include EL serving as an exogenous construct, PSRB as an endogenous construct, PsyCap functioning as an explanatory construct, and MID operating as a moderating construct. All study constructs were assessed using a five-point Likert scale that ranged from 1 (indicating strong disagreement) to 5 (indicating strong agreement). These scales relied on self-reporting, with employees offering their own perceptions regarding the study constructs [72]. To alleviate the potential influence of common method bias, responses were collected across three distinct time periods [70]. To counteract any effects of social desirability bias, the data collection process was meticulously designed to incorporate essential precautions [73]. Furthermore, certain measurement items were adapted and tailored to align with the hospital-specific setting of the present investigation [74].

## Ethical Leadership (EL)

The study utilized the 10-item Ethical Leadership Scale (ELS) developed by Brown et al. [1] to assess EL at T1. A sample item from the scale includes: "My supervisor engages in discussions about business ethics or values with employees." The Cronbach's alpha coefficient for EL was calculated as .919, indicating high internal consistency reliability.

## Pro-Social Rule Breaking (PSRB)

At T3, the study employed the 13-item scale developed by Dahling et al. [37] to assess PSRB. A sample item from the scale reads: "I break organizational rules to increase efficiency and cost

**Table 2. Total response rate.**

| Time Lags | Constructs Measured | Questionnaires Distributed | Responses Received | Percentage of Responses |
|---|---|---|---|---|
| T1: (1 April– 31 May, 2022) | EL, MID, Demographics | 900 | 751 | 83.44% |
| T2: (1 June– 31 July, 2022) | PsyCap, | 751 | 649 | 86.41% |
| T3: (1 August– 30 September, 2022) | PSRB | 649 | 591 | 91.06% |

**Table 3. Valid response rate.**

| No. of Questionnaires | Valid Response Rate |
|---|---|
| Total Questionnaires Distributed | 900 |
| Total Questionnaires Received | 591 |
| Response Rate of Total Questionnaires Received | 65.66%. |
| Questionnaires Rejected Due to Incomplete Information | 47 |
| Questionnaires Rejected Due to Invalid Response | 29 |
| Total No. of Valid Questionnaires | 515 |
| Valid Response Rate | 57.22%, |

savings." The Cronbach's alpha coefficient for PSRB was calculated as .937, indicating strong internal consistency reliability.

## Psychological Capital (PsyCap)

At T2, the study employed the 12-item short version of the PsyCap Questionnaire (PCQ-12) developed by Martínez and colleagues [75]. Originally, the PCQ consisted of 24 items (PCQ-24), first developed by Luthans and colleagues [20]. A sample item from the PsyCap scale included: (a) efficacy: "I am confident in my abilities to perform under pressure and challenging circumstances"; (b) resilience: "I remain resilient and ready to face difficulties at work"; (c) hope: "When I set goals and plan my work, I concentrate on achieving them"; and (d) optimism: "I believe that every problem at work has a positive aspect." The Cronbach's alpha coefficient for PsyCap was .937, indicating robust internal consistency reliability.

**Table 4. Sample characteristics.**

| Demographics | Frequency (n = 515) | Percentage |
|---|---|---|
| Gender | | |
| Male | 219 | 42.5 |
| Female | 296 | 57.5 |
| Marital Status | | |
| Single | 163 | 31.7 |
| Married | 352 | 68.3 |
| Age | | |
| 21–30 years | 123 | 23.9 |
| 31–40 years | 258 | 50.1 |
| 41–50 years | 110 | 21.4 |
| 51–60 years | 24 | 4.7 |
| Education | | |
| Bachelors | 214 | 41.6 |
| Masters | 221 | 42.9 |
| MS/MPhil | 80 | 15.5 |
| Ph.D. | - | - |
| Experience | | |
| 1–5 years | 230 | 44.7 |
| 6–10 years | 162 | 31.5 |
| 11 – 15 years | 78 | 15.1 |
| 16–20 years | 37 | 7.2 |
| >20 years | 8 | 1.6 |

**Table 5. Descriptive statistics.**

| Constructs | N | Missing | Min | Max | Mean | SD | Skewness | Kurtosis |
|---|---|---|---|---|---|---|---|---|
| EL | 515 | 0 | 1 | 5 | 3.962 | .816 | -2.024 | 4.196 |
| PSRB | 515 | 0 | 1 | 5 | 4.112 | .731 | -2.362 | 6.287 |
| PsyCap | 515 | 0 | 1 | 5 | 4.087 | .789 | -2.363 | 5.918 |
| MID | 515 | 0 | 1 | 5 | 3.003 | 1.411 | .405 | -1.472 |

EL: Ethical Leadership; MID: Moral Identity; PC: Psychological Capital; PSRB: Pro-Social Rule Breaking

### Moral Identity (MID)

At T1, the study utilized the 5-item MID (Internalization) scale developed by Aquino and Reed [24]. A sample item from the MID scale was: "I strongly aspire to possess the aforementioned qualities." The Cronbach's alpha coefficient for Moral Identity was .915, indicating high internal consistency reliability.

### Data analysis

The data analysis for this study utilized SPSS and PLS-SEM with Smart PLS 4 software. SPSS was employed for tasks such as data entry, coding, and initial data screening. Descriptive and frequency statistics were computed to explore the dataset, and various tests including assessments for normality and common method bias were conducted to evaluate data distribution and potential biases. The Harman Single-Factor test revealed that only 25.915% of the variance was explained, which fell below the critical threshold of 50%, indicating that common method bias was not a significant concern in the dataset.

Following this, PLS-SEM with Smart PLS was employed to establish and validate the measurement model, structural relationships, and test hypotheses. PLS-SEM facilitated the simultaneous examination of mediation and moderation effects within the theoretical framework. Additionally, the study evaluated the framework's predictive power by comparing it with the observed data [76–78].

### Ethical considerations

The study adhered to rigorous ethical principles, obtaining authorization from both public and private hospitals with the top management support. Strict ethical protocols safeguarded participant rights and confidentiality. The questionnaires were accompanied by a cover letter for informed consent and confidentiality assurance while highlighting the voluntary nature of participation and the absence of a need for personal or institutional identification. A systematic coding system was applied to align responses across three phases of the study. Ethical approval of the study was granted by the university review board. The study maintained and upheld international ethical standards and prioritized participants' well-being and privacy.

## Results

### Measurement model

Before assessing the structural model and hypotheses, the reliability and validity of the measurement model (outer model) were evaluated [76–78]. The depiction of the measurement model is presented in Fig 2.

To assess the internal consistency and reliability of the indicators, we conducted an analysis of outer loadings (OL), Cronbach's alpha (α), and composite reliability (CR). Except for EL 7,

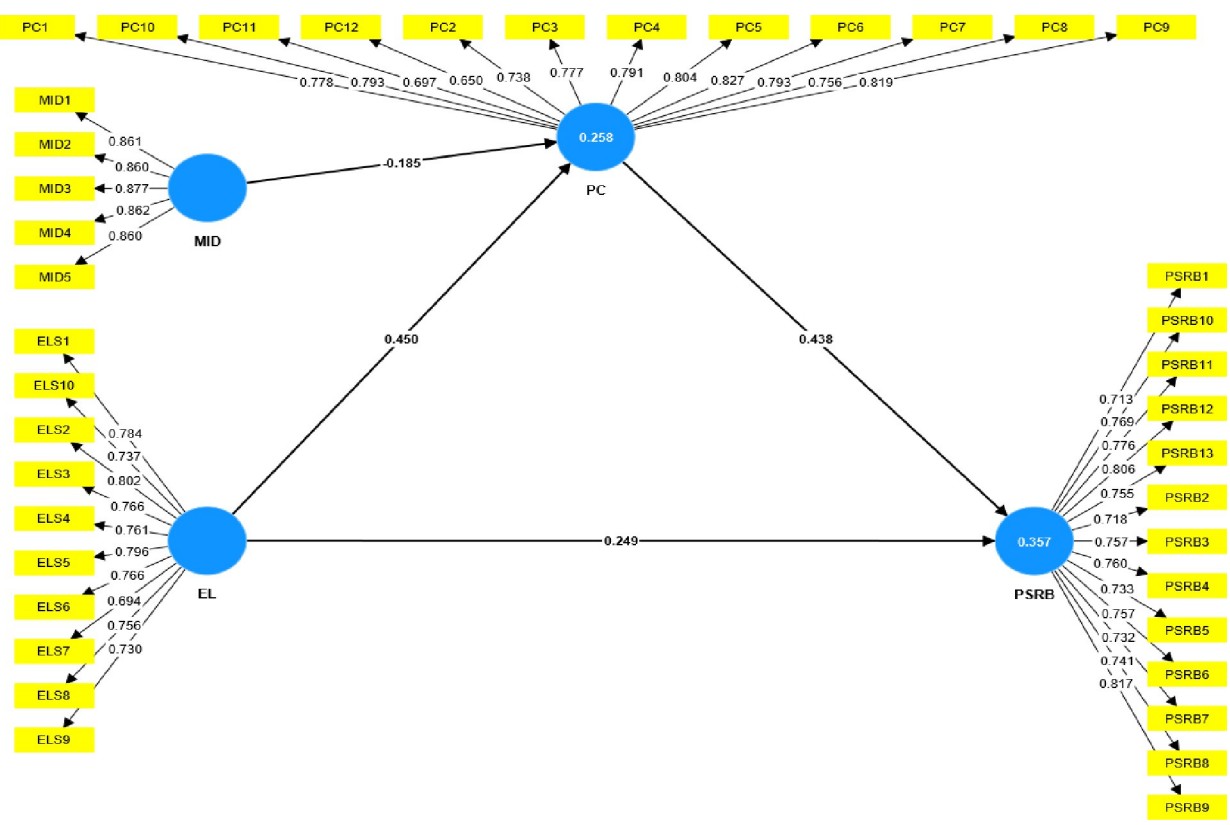

**Fig 2. Measurement model.**

PC 11, and PC 12, all items of the study variables exhibited OLs exceeding 0.708, accounting for over 50% of the variance in the indicators. Consequently, no items were excluded from any of the study variables [78]. Table 6 reflects the specific OLs of all study constructs. Additionally, both α and CR surpassed the threshold of 0.708 for all study variables, affirming the measurements' reliability, as depicted in Table 6.

Convergent validity of the study variables was assessed by examination of the Average Variance Extracted (AVE). The AVE for all study variables exceeded the threshold of 0.50 while staying below 0.85, affirming the convergent validity of the constructs [77], as outlined in Table 6.

Although EL7, PC11, and PC12 fall below the 0.70 threshold, they still have acceptable factor loadings above 0.60. Moreover, other reliability and validity measures support the overall robustness of our measurement model. The α and CR scores for all study constructs exceeded 0.708, indicating good internal consistency. Additionally, the AVE values for each construct exceeded 0.50, demonstrating good convergent validity. Therefore, we retained these items within their respective study constructs [76].

To establish the discriminant validity of the constructs, we employed both the Fornell and Larcker criterion and the Heterotrait Monotrait Ratio (HTMT). The AVE for all variables surpassed their correlations with other study variables. As a result, the Fornell and Larcker criterion confirmed the discriminant validity of the variables [77], as demonstrated in Table 7.

HTMT is a statistical measure utilized for assessment of the discriminant validity of variables within a structural equation modeling (SEM) framework. It calculates the ratio of the average correlations between variables to the average correlations within variables, aiding in

**Table 6. Construct reliability and convergent validity.**

| Measurements | OL | VIF | α | CR | AVE |
|---|---|---|---|---|---|
| EL | | | 0.919 | 0.932 | 0.580 |
| EL1 | 0.784 | 2.248 | | | |
| EL2 | 0.802 | 2.392 | | | |
| EL3 | 0.766 | 2.478 | | | |
| EL4 | 0.761 | 2.471 | | | |
| EL5 | 0.796 | 2.475 | | | |
| EL6 | 0.766 | 2.237 | | | |
| EL7 | 0.694 | 2.684 | | | |
| EL8 | 0.756 | 2.891 | | | |
| EL9 | 0.73 | 2.919 | | | |
| EL10 | 0.737 | 2.791 | | | |
| MID | | | 0.915 | 0.936 | 0.745 |
| MID1 | 0.861 | 2.689 | | | |
| MID2 | 0.86 | 2.594 | | | |
| MID3 | 0.877 | 2.93 | | | |
| MID4 | 0.862 | 2.584 | | | |
| MID5 | 0.86 | 2.371 | | | |
| PC | | | 0.937 | 0.945 | 0.592 |
| PC1 | 0.778 | 2.457 | | | |
| PC2 | 0.738 | 2.283 | | | |
| PC3 | 0.777 | 2.319 | | | |
| PC4 | 0.791 | 2.642 | | | |
| PC5 | 0.804 | 2.664 | | | |
| PC6 | 0.827 | 2.731 | | | |
| PC7 | 0.793 | 2.552 | | | |
| PC8 | 0.756 | 2.202 | | | |
| PC9 | 0.819 | 2.704 | | | |
| PC10 | 0.793 | 2.418 | | | |
| PC11 | 0.697 | 2.214 | | | |
| PC12 | 0.65 | 1.995 | | | |
| PSRB | | | 0.938 | 0.946 | 0.573 |
| PSRB1 | 0.713 | 2.04 | | | |
| PSRB2 | 0.718 | 2 | | | |
| PSRB3 | 0.757 | 2.069 | | | |
| PSRB4 | 0.76 | 2.135 | | | |
| PSRB5 | 0.733 | 2.153 | | | |
| PSRB6 | 0.757 | 2.19 | | | |
| PSRB7 | 0.732 | 1.984 | | | |
| PSRB8 | 0.741 | 2.16 | | | |
| PSRB9 | 0.817 | 2.713 | | | |
| PSRB10 | 0.769 | 2.214 | | | |
| PSRB11 | 0.776 | 2.297 | | | |
| PSRB12 | 0.806 | 2.574 | | | |
| PSRB13 | 0.755 | 2.04 | | | |

EL: Ethical Leadership; MID: Moral Identity; PC: Psychological Capital; PSRB: Pro-Social Rule Breaking

**Table 7. Fornell-Larcker criterion.**

| Constructs | EL | MID | PC | PSRB |
|---|---|---|---|---|
| EL | 0.76 | | | |
| MID | -0.124 | 0.864 | | |
| PC | 0.473 | -0.241 | 0.77 | |
| PSRB | 0.456 | -0.251 | 0.556 | 0.757 |

EL: Ethical Leadership; MID: Moral Identity; PC: Psychological Capital; PSRB: Pro-Social Rule Breaking

determining whether the variables under examination are empirically distinct. In this study, the correlations between the different study constructs were identified to be less than 0.85, thereby affirming the discriminant validity of the constructs as indicated by HTMT [78]. This is illustrated in Table 8.

## Structural model

The structural model, referred to as the inner model, was evaluated using various metrics such as multicollinearity, coefficient of determination (R2), effect size (F2), predictive relevance (Q2), and the statistical significance and practical relevance of the path coefficients [77]. The structure model is represented in Fig 3.

To confirm the absence of collinearity among the study variables, we examined the Variance Inflation Factor (VIF) values of all the indicators. The VIF values for all the indicators were determined to be below 3.0, signifying the absence of collinearity among the study variables [78], as presented in Table 6 above.

The in-sample explanatory power of the model was evaluated using the coefficient of determination ($R^2$). The $R^2$ assesses the model's predictive accuracy by measuring the correlation between actual and expected values of the endogenous construct. $R^2$ values range from 0 to 1, with higher values indicating a stronger explanatory power. $R^2$ values of 0.75 are considered as substantial, 0.50 as moderate, and 0.25 as weak [77]. Specifically, the $R^2$ value for PSRB was 0.423, indicating that EL explained 42.3% of the variance in PSRB. This suggests that EL can moderately explain the variance in PSRB, highlighting its significance while recognizing the likely influence of other factors on this behavior.

The effect size (F2) measured the extent of variance introduced by a singular independent variable in the dependent variable. The $F^2$ quantifies the impact of removing a specific independent variable on the $R^2$ of a dependent variable, signifying the independent variable's contribution. $F^2$ values are considered large at 0.35, medium at 0.15, and weak at 0.02 [77]. For the relationship between EL and PSRB, the effect size was 0.038, indicating that EL's contribution to the variance in PSRB is relatively modest.

**Table 8. Discriminant validity (HTMT ratio).**

| Constructs | EL | MID | PC | PSRB |
|---|---|---|---|---|
| EL | | | | |
| MID | 0.133 | | | |
| PC | 0.504 | 0.259 | | |
| PSRB | 0.487 | 0.268 | 0.59 | |

EL: Ethical Leadership; MID: Moral Identity; PC: Psychological Capital; PSRB: Pro-Social Rule Breaking

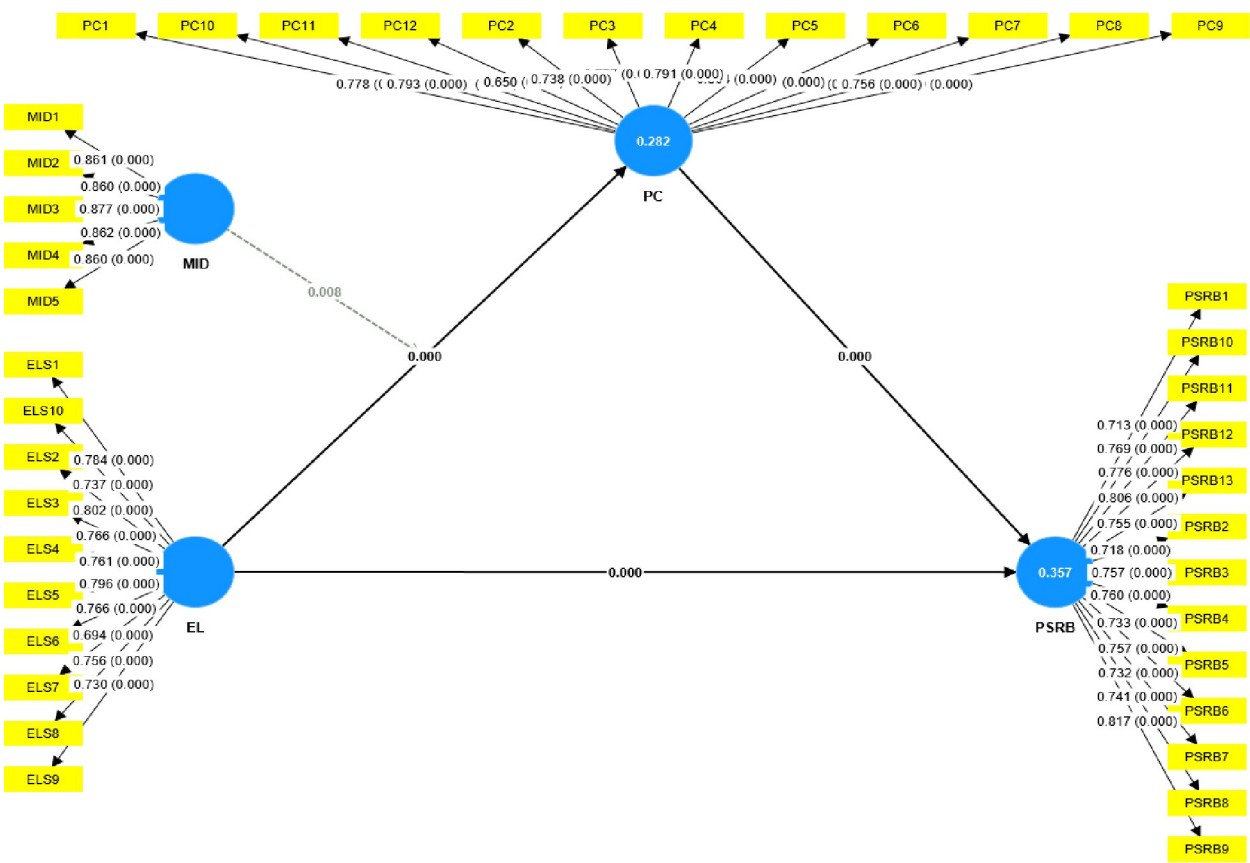

**Fig 3. Structural model.**

To assess the predictive precision of the model, Q2 was computed employing the blindfolding technique. $Q^2$ evaluates the model's ability to predict beyond the sample through out-of-sample prediction in a blindfolding process. $Q^2$ values above zero are meaningful for predictive accuracy, with values exceeding 0 indicating small, 0.25 as medium, and 0.50 as large predictive relevance for the structural model [76]. $Q^2$ predict value for PSRB was 0.242, suggesting that EL has a medium level of predictive relevance for PSRB [77].

Overall, the model has moderate explanatory relevance, as EL explains a substantial part of PSRB variance. Predictive relevance is also moderate, meaning EL can moderately predict PSRB variations. The F2 value indicates EL's impact on PSRB is relatively modest, highlighting the importance of considering other factors as shown in Table 9.

**Table 9. Explanatory and predictive relevance of the model.**

| Parameters | EL > PSRB |
|---|---|
| $R^2$ | 0.423 |
| $F^2$ | 0.038 |
| $Q^2$ | 0.242 |

EL: Ethical Leadership; PSRB: Pro-Social Rule Breaking; $R^2$: Coefficient of Determination; $F^2$: Effect Size; $Q^2$: Predictive Relevance

**Table 10. Hypotheses testing.**

| Hypotheses | Relationships | β | t Values | p Values | LLCI | ULCI | Results |
|---|---|---|---|---|---|---|---|
| H-1 | EL > PSRB | 0.154 | 3.141 | 0.002 | 0.062 | 0.254 | Not Supported |
| H-2 | EL > PC > PSRB | 0.128 | 3.887 | 0.001 | 0.071 | 0.202 | Supported |
| H-3 | MID X EL > PC | 0.16 | 2.58 | 0.01 | 0.032 | 0.274 | Supported |

EL: Ethical Leadership; PSRB: Pro-Social Rule Breaking; PC: Psychological Capital; MID: Moral Identity

## Hypotheses testing

The hypotheses were examined for the direct, mediated and moderated relationships employing PLS SEM (Smart PLS) [76–78]. The results can be found in Table 10 below.

Hypothesis1 postulated that there was a negative link between EL and employees' PSRB in the organization. However, the finding showed that EL was significantly and positively related to the employees' PSRB (β = 0.154; t = 3.141; p = 0.002). The $F^2$ value reflected that EL was positively linked to the employees' PSRB ($F^2$ = 0.038).The direct effect had no zero in between LLCI and ULCI at 95% CI (LLCI = 0.062; ULCI = 0.254). The relationship between EL and employees' PSRB was statistically significant, but contrary to hypothesis 1. Therefore hypothesis1 was not supported in the expected direction.

Hypothesis 2 proposed that the employees' PsyCap mediated the linkage between EL and employees' PSRB in the organization. The finding revealed that specific indirect effect of PsyCap between EL and PSRB was significant (β = 0.128; t = 3.887; p = 0.001). The specific indirect effect had also no zero in between LLCI and ULCI at 95% CI (LLCI = 0.071; ULCI = 0.202). Therefore, hypothesis 2 received confirmation.

Hypothesis 3 postulated that the employees' MID moderated the link between EL and employees' PsyCap such that the link was stronger for employees with higher MID. The interaction term of MID X EL was added to the direct relationship of EL and PsyCap. The results showed that the direct effect was statistically still significant even in presence of interaction term of MID X EL (β = 0.16; t = 2.58; p = 0.01; LLCI = 0.032; ULCI = 0.274). The interaction plot represented that MID at + 1 SD was rising upward steeper than MID at– 1 SD. This showed that MID strengthened the link between EL and employees' PsyCap more for employees with higher MID. Therefore hypothesis 3 was supported. The interaction plot was shown in the Fig 4.

## Discussion

The study's results provide valuable insights into the relationships between EL, employees' PsyCap, MID, and PSRB within the organizational context. Hypothesis 1 postulated that there was a negative link between EL and employees' PSRB. However, the findings contradicted this hypothesis, as they unveiled a statistically significant and positive association between EL and employees' PSRB (β = 0.154; t = 3.141; p = 0.002; LLCI = 0.062; ULCI = 0.254). Recent research conducted in developing nations such as Iran, Turkey, and Pakistan has identified a negative correlation between EL and employees' engagement in workplace deviant behaviors [79–81]. Moreover, this finding also contradicts recent literature that has suggested a negative link between EL and employees' constructive deviance behaviors, such as UPB [82, 83].

The R2 value of 0.423 indicates that EL moderately explains PSRB variance, highlighting its significance while acknowledging the influence of other factors. EL is a significant factor in understanding PSRB but doesn't explain all variance. Organizations should recognize the role of additional factors in PSRB and the importance of promoting EL practices to positively impact PSRB.

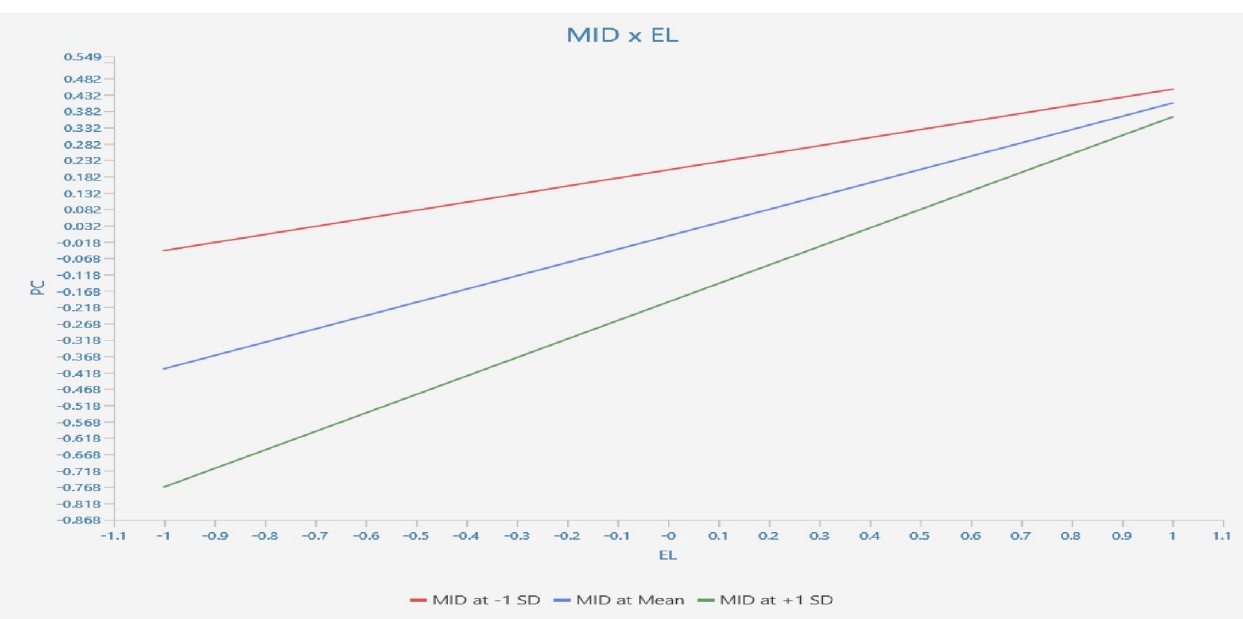

**Fig 4. Moderation graph.**

An F2 value of 0.038 suggests that EL's impact on PSRB is relatively modest, indicating that EL alone does not have a very strong explanatory effect. Other factors likely play a role in explaining the variance in PSRB. Practically, organizations should understand that promoting EL alone may not be sufficient to substantially influence PSRB, necessitating consideration of additional strategies or factors.

With a Q2 value of 0.242, EL has a medium level of predictive relevance for PSRB, meaning it is moderately effective at predicting variations in PSRB. EL is a meaningful factor in understanding PSRB, but other variables also play a role. Organizations should emphasize promoting EL to manage PSRB while considering additional variables and strategies.

This unexpected positive correlation in the healthcare context can be attributed to the unique demands and ethical priorities of the industry, where flexibility in rule adherence is sometimes seen as necessary for patient welfare. Factors such as the need for rule flexibility, organizational culture, peer and leadership dynamics, and ethical considerations contribute to nursing staff engaging in PSRB despite the presence of EL. The emphasis on patient well-being and the complexity of healthcare systems likely lead to increased instances of PSRB [9, 84]. Recognizing these contextual and organizational factors is crucial for interpreting the study's results.

This counterintuitive finding highlights the variability in the impact of EL across different settings, emphasizing the significance of contextual and organizational factors in understanding its effects. While it does not fundamentally challenge the concept of EL, it highlights the significance of considering how employees perceive and respond to EL in diverse contexts [8, 40, 43]. This points to the necessity of acknowledging the unique dynamics of different industries and cultures when assessing the impact of EL.

Our research focuses on examining pro-social intentions and behaviors, which may sometimes require departing from established norms or rules. Such behavior can involve bending rules to help others or achieve organizational objectives, particularly when employees believe these actions serve the greater good of the organization [11, 35, 85].

Hypothesis 2 postulated that employees' PsyCap mediated the link between EL and employees' PSRB. The results supported this hypothesis, as the specific indirect effect of PsyCap between EL and PSRB was significant ($\beta$ = 0.128; t = 3.887; p = 0.001; LLCI = 0.071; ULCI = 0.202). This finding underscores the role of employees' psychological resources, specifically PsyCap, in explaining how EL influences their PSRB. It aligns with SCT, highlighting the role of cognitive and psychological processes in translating leadership into employee actions.

The results supported this hypothesis, aligning seamlessly with previous research that consistently demonstrates PsyCap as an explanatory mechanism between leadership and employee behaviors [50]. This finding also resonates with a study conducted in Pakistan, where researchers found that PsyCap and goal congruence help explain the positive impact of EL on in-role job performance [57]. These results highlight the critical role of PsyCap in elucidating the connection between EL and employees' PSRB within the organization.

By considering employees' PsyCap as a mediating factor, this study contributes valuable insights to the existing body of knowledge, enhancing our understanding of how EL influences PSRB through the lens of PsyCap. This provides a fresh and insightful perspective on the intricate relationship between EL and PSRB, emphasizing the mediating role of PsyCap.

These findings underscore the importance of promoting PsyCap in the workplace, as it significantly shapes the organization's pro-social environment. They highlight the need for policies aimed at fostering pro-social behaviors. Furthermore, the results emphasize the critical role of EL in shaping employees' psychological processes and pro-social behaviors, underscoring the necessity for organizations to actively endorse and support EL practices. This involves establishing explicit policies and guidelines for leaders at all organizational levels to ensure a consistent and supportive leadership approach.

Hypothesis 3 suggested employees' MID moderated the link between EL and PsyCap, with a stronger link for employees with higher MID levels. The findings supported this hypothesis, demonstrating that the direct effect between EL and PsyCap was significant even in the presence of the interaction term of MID X EL ($\beta$ = 0.16; t = 2.58; p = 0.01; LLCI = 0.032; ULCI = 0.274). The interaction plot indicated that MID at +1 SD had a steeper upward slope than MID at -1 SD, suggesting that MID enhanced the link between EL and employees' PsyCap more significantly for employees with higher MID levels. The result underscores the role of individual differences, particularly MID, when investigating how EL influences employees' cognitive and psychological processes like PsyCap.

The results supported this hypothesis, aligning with previous studies that demonstrate more pronounced moderating influence of MID in the relationship between leadership and various psychological mechanisms and behaviors for individuals with elevated MID levels [61]. This finding is also consistent with research conducted in Iraq, which showed that EL has a more substantial positive impact on employees' ethical conduct when they possess both a strong MID and high self-control [21]. This underscores the significance of considering both leaders' and subordinates' attributes to optimize the benefits of EL in an organizational context.

Additionally, the study provides compelling evidence that individuals with higher MID exhibit heightened responsiveness to EL behaviors, leading to a more substantial enhancement of their PsyCap. This finding not only enriches the existing literature but also highlights the significance of considering employees' MID when investigating the influence of EL on employees' cognitive and psychological processes such as PsyCap.

These findings emphasize the importance of organizational focus on leadership selection, development, and EL training. Organizations should assess moral values and ethical principles when appointing leaders to ensure alignment with the organization's ethics. Leadership development programs should nurture MID, preparing leaders to promote ethical practices

effectively. Furthermore, these results underscore the need for adaptable leadership approaches that consider employees' varying moral values, tailoring leadership styles to enhance EL's impact on PsyCap.

## Theoretical contributions

This study makes significant theoretical contributions to the fields of leadership and organizational behavior. Firstly, it reinforces the fundamental premise of SCT by empirically demonstrating that external environmental factors, exemplified by EL, significantly influence individuals' cognitive processes, such as PsyCap, and behaviors, including PSRB [25]. This empirical validation underscores the pivotal role of EL in shaping employees' PsyCap and PSRB, challenging traditional assumptions that portray EL as inherently and exclusively positive [45]. By doing so, it highlights the necessity for a more nuanced understanding of EL, one that acknowledges its potential to encourage PSRB under certain conditions, thereby revealing the complex dynamics of leadership within organizational settings.

Secondly, the study introduces PsyCap as a critical mediating mechanism within the SCT framework. By establishing PsyCap as a mediator between EL and PSRB, it enriches our comprehension of how EL's influence on cognitive processes translates into specific employee behaviors. This finding aligns with SCT's emphasis on cognitive processes and self-regulation, adding depth to its application in understanding the impact of leadership on cognitive resources [28]. Moreover, the study brings new complexities to the PsyCap literature by uncovering potential adverse outcomes, such as employees' involvement in PSRB. This challenges the prevalent notion that PsyCap uniformly produces benefits for both individuals and organizations, highlighting the significance of considering the underlying psychological mechanisms that may lead to unethical behaviors.

Thirdly, the study emphasizes the moderating role of MID in the link between EL and employees' PsyCap, showing that this interaction is more pronounced for individuals with higher MID levels. This finding expands SCT's understanding by illustrating the bidirectional influences between individual differences (MID) and external environmental factors (EL) [27]. It demonstrates how individual differences can amplify or diminish the effects of external factors like EL, thereby enriching the conceptual framework of SCT. Additionally, it enhances our understanding of the boundary conditions of the EL-PsyCap link, stressing the significance of individual differences, particularly MID, in influencing how EL affects employees' cognitive and psychological processes [21].

Lastly, the study adopts a holistic approach by integrating leadership, individual differences, psychological processes, and behavior. By incorporating EL, MID, PsyCap, and PSRB, it addresses a notable gap in SCT's application by explaining real-world workplace behaviors, especially within non-Western developing nations such as Pakistan. SCT has traditionally focused on cognitive and social cognitive processes primarily in developed nations [25, 27, 31]. This investigation extends the applicability of SCT to non-Western developing countries, demonstrating how EL's influence on cognitive processes directly affects employee behavior in these regions. This comprehensive perspective acknowledges the intricate interplay between leadership styles, individual characteristics, cognitive processes, and resulting behaviors, thereby enriching SCT's theoretical framework.

Overall, these theoretical contributions significantly expand our understanding of EL and its effects on employees' behavior. They offer valuable insights for future research and provide practical guidance for leadership practices across diverse organizational contexts, including non-Western developing countries. By challenging conventional assumptions and highlighting the complexities of EL in contemporary workplaces, this study underscores the need for a

more sophisticated and context-sensitive approach to understanding and implementing ethical leadership.

## Managerial implications

The policy recommendations derived from our study offer a practical framework for translating research findings into actionable strategies. Given the unique challenges of different industries, such as balancing ethics and patient welfare in healthcare, we emphasize the importance of industry-specific leadership training. This training should address the nuances of EL within the specific context, helping leaders navigate the complexities of rule flexibility and ethical decision-making. Establishing clear, industry-specific ethical guidelines can clarify when rule flexibility is appropriate, and leaders should model these values consistently. Additionally, organizations should address the dynamics of peer and leadership interactions to foster a culture that aligns with EL principles and meets industry-specific requirements.

Our study also highlights the significance of developing employees' PsyCap through targeted training initiatives aimed at enhancing self-efficacy, hope, optimism, and resilience. Leadership programs should integrate EL training and promote policies that encourage active participation in these programs. Incorporating EL and PsyCap development into performance evaluations is crucial, making these aspects integral to leadership roles. Furthermore, comprehensive employee well-being policies can support the development of PsyCap, contributing to a more resilient and positive workforce.

Recognizing and supporting employees with higher levels of MID is another key recommendation. Public acknowledgment of these individuals can serve as a powerful motivator. Creating opportunities for mentorship, establishing peer support groups, and forming ethical committees can foster a sense of community and shared ethical standards. Implementing feedback systems for ethical concerns can help refine organizational ethics continuously. Additionally, adopting flexible leadership approaches that accommodate varying levels of MID can enhance the effectiveness of EL practices across the organization.

The study's findings indicate that while EL explains a significant portion of the variance in PSRB, its overall predictive power is moderate. This suggests that EL is an important factor but not the sole determinant of PSRB. Organizations should consider these findings when addressing PSRB, recognizing that other factors also play a crucial role. The moderate explanatory relevance of the model highlights the need to incorporate additional elements into strategies for mitigating PSRB, ensuring a comprehensive approach to ethical leadership and employee behavior management.

## Limitations and future research directions

Based on the results and limitations of this study, several avenues for future research can be identified. First, the study focused on nursing staff in the health sector in Pakistan. This limits the generalizability of the findings to other industries, job roles, and cultural contexts. Future research could replicate the study in different organizational settings and cultures to enhance the external validity of the findings.

Second, the study employed a time-lagged design, which, despite its strengths, still had a cross-sectional nature. This limitation restricts the ability to establish causal relationships between variables. To capture the dynamic nature of EL, employees' behavior, and the mediating and moderating factors under investigation, future research could consider employing longitudinal or experimental designs.

Third, while the study identified PsyCap as a mediator, there may be other variables at play. Future research could explore additional mechanisms, such as psychological empowerment

and organizational identification, to better understand the link between EL and employee outcomes. Moreover, future research can focus on a detailed examination of individual components of PsyCap, including self-efficacy, optimism, hope, and resilience to investigate how each of these components mediates the relationship between EL and PSRB. These can provide a more nuanced understanding of their specific roles in this context.

Fourth, future research could investigate the specific contributions of individuals with higher MID levels to the overall organizational climate. This could include examining how they influence ethical climate, moral values, and the ethical decision-making climate within an organization.

Fifthly, as an avenue for future research, it is recommended to broaden the investigation of EL beyond its impact on PSRB. Future studies should explore EL's influence on a spectrum of constructive deviant behaviors like employees' UPB and whistleblowing.

Sixthly, it is highly encouraged for forthcoming research to explore the connections between various moral leadership styles and employees' PSRB. In addition to EL, examining leadership styles such as authentic leadership, and servant leadership is considered essential. This will provide a more comprehensive understanding of how various moral leadership styles influence employees' PSRB.

Seventhly, the model provides valuable insights into the relationship between EL and PSRB, but it acknowledges the influence of other variables. To enhance the model's predictive relevance, researchers may need to explore additional factors that contribute to PSRB.

Finally, the study mentioned the limited research on the impact of EL on employees' PSRB, particularly in non-Western cultures. Future research could conduct comparative studies between Western and non-Western cultures to explore potential cultural differences in the effects of EL.

## Conclusion

The unexpected positive relationship between EL and PSRB behavior in our study emphasizes the need to reevaluate traditional notions of EL. It highlights the importance of contextual factors, such as industry-specific challenges and norms. To promote ethical behavior, organizations should prioritize tailored leadership training that considers these unique contexts. Developing employees' PsyCap is crucial, and recognizing individuals with strong MIDs can serve as a motivation. Organizations should focus on aligning EL with contextual needs to foster an ethical culture within the workplace. These findings offer valuable insights for organizations seeking to promote ethical behavior and leadership in their specific contexts.

## Supporting information

**S1 Dataset.**
(SAV)

## Author Contributions

**Conceptualization:** Mushtaq Ahmed, Muhammad Ishfaq Khan.

**Data curation:** Mushtaq Ahmed.

**Formal analysis:** Mushtaq Ahmed.

**Methodology:** Mushtaq Ahmed.

**Software:** Mushtaq Ahmed.

**Supervision:** Muhammad Ishfaq Khan.

**Writing – original draft:** Mushtaq Ahmed.

**Writing – review & editing:** Muhammad Ishfaq Khan.

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
