## [Decision Letter · Decision Letter 0]

22 Oct 2023

PONE-D-23-30966Unpacking the Paradoxical Impact of Ethical Leadership on Employees’ Pro-Social Rule-Breaking Behavior: The Interplay of Employees’ Psychological Capital and Moral IdentityPLOS ONE

Dear Dr. Ahmed,

Thank you for submitting your manuscript to PLOS ONE. After careful consideration, we feel that it has merit but does not fully meet PLOS ONE’s publication criteria as it currently stands. Therefore, we invite you to submit a revised version of the manuscript that addresses the points raised during the review process.

ACADEMIC EDITOR: Please address all the comments and suggestions given by the reviewers.==============================

We look forward to receiving your revised manuscript.

Kind regards,

Kashif Ali, PH.D

Academic Editor

PLOS ONE

Journal Requirements:

6. Please upload a copy of Figure 4, to which you refer in your text on page 26. If the figure is no longer to be included as part of the submission please remove all reference to it within the text.

Reviewers' comments:

Reviewer's Responses to Questions

**Comments to the Author**

1. Is the manuscript technically sound, and do the data support the conclusions?

Reviewer #1: Yes

Reviewer #2: Yes

Reviewer #3: Partly

Reviewer #4: Yes

Reviewer #5: No

2. Has the statistical analysis been performed appropriately and rigorously? 

Reviewer #1: No

Reviewer #2: Yes

Reviewer #3: No

Reviewer #4: Yes

Reviewer #5: Yes

3. Have the authors made all data underlying the findings in their manuscript fully available?

Reviewer #1: No

Reviewer #2: Yes

Reviewer #3: No

Reviewer #4: Yes

Reviewer #5: Yes

4. Is the manuscript presented in an intelligible fashion and written in standard English?

Reviewer #1: Yes

Reviewer #2: Yes

Reviewer #3: Yes

Reviewer #4: Yes

Reviewer #5: No

5. Review Comments to the Author

Reviewer #1: The paper shows promise, but there are several minor corrections and improvements that could enhance its quality. Here are specific recommendations:

1. The introduction (Pages 6-7): I suggest writing the full terms before the abbreviations (e.g., Psychological Capital (PsyCap), Moral Identity (MID), and Social Cognitive Theory (SCT)) to improve clarity and understanding especially in the first paragraph.

2. Pro-Social Rule-Breaking Behaviour in Health Industries: Please provide a more detailed explanation of how employees in the health industry practice Pro-Social Rule-Breaking Behaviour. This will help readers better grasp the context and relevance of your study.

3. Choice of Research Design: far away of the limitations you have mentioned, Clarify the rationale behind employing a time-lagged cross-sectional design with three data collection waves (T1, T2, and T3). Why do you use this strategy which could cause CMB rather than minimize it.

4. Data Collection Details: Provide more information about data collection from each hospital, including response rates for each, which is crucial for understanding the representativeness of your sample.

5. You mentioned conducting a Common Method Bias (CMB) test, please included the results in the paper for transparency and to support your research validity.

6. Factor Loadings (Items EL7, PC11, PC12) are below 0.70. Explain your reasoning for retaining these items in the analysis, as they could impact the reliability and validity of your results.

7. Include the results of Fornell-Larcker criteria in your paper.

8. In page 23, you mentioned VIF values, but it appears you are referring to variables rather than indicators. Please include the VIF values for the indicators to clarify this point.

9. F2 values is 0.038 which indicate to small effect size and not large effect size. Moreover, Q2 is 0.242 which is weak and not large. Ensure the interpretations accurately reflect the analysis outputs.

Addressing these points will enhance the clarity, validity, and overall quality of your paper. Your efforts are commendable, and these improvements will strengthen your research.

Reviewer #2: General Comments:

Overall, the study is well-executed, providing clear insights into the influence of ethical leadership on employee behavior. Consider the following points, however, to improve the paper's quality:

Abstract

It might be beneficial to finish the abstract by discussing the larger significance of these results for organizational leadership and suggesting future research alternatives in this field. This would bring the narrative to a close and offer the reader with a clear takeaway message.

Introduction

In general, the introduction has useful information, but it might be improved by simplifying, giving precise terminology (e.g., PsyCap, MID), and specifying particular study aims or questions. This will provide the study with a more focused and ordered foundation.

Theory and Hypotheses Development

1. Rather than using the word "paradoxical relationship" without explanation, the text should explain why the observed positive association between ethical leadership and PSRB is regarded as paradoxical.

2. The statement "shedding light on the dynamic nature of ethical leadership" is a little ambiguous. A more specific explanation of how the findings contribute to our knowledge of ethical leadership dynamics would be helpful in this area.

3. It is critical to address any potential critiques or limitations connected with the selected theoretical framework when introducing it. This indicates a sophisticated comprehension of the theoretical perspective.

Methods

Population and Sampling

1. Rationale for Sample Size: Explain why a higher sample size (500) was chosen, even when the GPower calculator indicated a minimum of 150. The methodological basis would be better if the reasons behind this decision were clarified.

2. Data Collection Procedure: Include a section explaining the quality control procedures used during data collection, including the efforts required in order to ensure accurate and reliable responses.

3. Measurement Validity and Reliability: Add a paragraph outlining the measurement instrument's reliability and validity, as well as references to any relevant validation studies.

4. Ethical Considerations: Since you stated getting permission from the hospitals, consider including a statement about ethical approval and any actions taken to protect the confidentiality of participants and informed consent. Include a section addressing the study's ethical issues, such as permission from appropriate ethics boards and safeguards in place to preserve the confidentiality of participants and rights.

Results

1. Structural Model Evaluation:

Provide a brief analysis of the R2, F2, and Q2 values, emphasizing their significance for the model's explanatory and predictive abilities.

Discussion

1. Interpretation of Contradictory Results: Are there contextual factors or specific organizational dynamics that might contribute to this unexpected outcome?

2. Interpretation of the Counterintuitive Finding:

3. Implications for EL Perception: Could this finding imply that workers understand ethical leadership differently than previously thought?

4. Distinguishing Pro-Social Intentions:

• Distinguishing Intentions from Actions: Add a brief remark to highlight that the research is concentrating on pro-social intentions and behaviors, even if they require breaking specific norms.

5. Mediation Role of PsyCap:

• Potential Mechanisms: Investigate potential mechanisms or specific components of PsyCap that may be particularly relevant in mediating the link between EL and PSRB.

6. Discussion of Moderating Effect:

• Further Implications of MID: How might individuals with higher MID levels contribute differently to the overall organizational climate?

Reviewer #3: 1. The topic is quite interesting, but I am sharing a few observations which may be incorporated in the next version of the manuscript.

2. Why the use of SPSS? Please clarify because descriptive statistics can also be done through PLS-SEM.

3. Do mention the original scale of the following construct EL, MID, PC, PSRB

4. Specify Outer VIF Values and Lacker criteria

5. Please recheck H1 EL > PSRB P value is 0.002, but the author mentioned it is not supported, which is incorrect.

6. Page no 21 Table 3: Items under EL (10), MID (05), PC (12), PSRB (13), but in Figure 2, Items are mismatched.

7. You may please refer to the following relevant references for your further perusal:

https://doi.org/10.1108/QRFM-04-2017-0028

https://doi.org/10.1108/QRFM-07-2018-0081

https://doi.org/10.1016/j.jclepro.2023.138441

https://doi.org/10.1111/ijcs.12850

Reviewer #4: 1. The article is about ethical leadership and found to be good. The article needs to be strengthened and introduced well on various aspects of ethical leadership.

2. The introduction needs to be focused on relevant literature, currently it seems to verbose and difficult to follow.

3. The sampling have to be explained and justified in a manner it could be replicated in other places and defined well.

4. The article provide results in table but not highlighted the relevant results having policy implication in the text.

5. The discussion section needs to be improved by including studies of developing countries instead of skewed representation of work.

6. The study needs a focused and actionable conclusion on ethical leadership by interpreting the data and results.

7. Overall, the study is conveying a message and need to be improved.

Reviewer #5: Thank you very much for giving me with an opportunity to review this interesting study titled: Unpacking the Paradoxical Impact of Ethical Leadership on Employees’ Pro-Social Rule-Breaking Behavior: The Interplay of Employees’ Psychological Capital and Moral Identity.

There are few concerns that needs major revision before publication is made with the esteemed journal.

-In the introduction, please discuss your gap more prominently. It is suggested to provide strong logic of the mediating and moderating inclusions instead of claiming that this study is the first to test these relationships.

-For the methods sections-The data was collected using English as the medium of instruction. My very basic concern is that most of the nurses are not well-educated in Pakistan. How come the provided questionnaire can yield a very productive and non-biased outcome?

-Along with that, the key concepts should be defined and operationalized from the start. Instead of providing your definitions in the hypotheses provide them in the introduction for more convincing flow of this section.

-Second, the authors have provided a statement with university degrees. Okay we for one moment take this point in consideration as valid. My again a very basic question is that how many nurses are well-educated with nursing degrees?

-Third, the authors have stated that they have collected their data in 3 waves with 751 in the first round, 649 questionnaires in the second round and finally 551 nurses in the third round. How the authors have ensured that nurses have university degrees. Where most of the nurses are not university qualified.

-What was the sampling technique which is employed. Was it probability or non-probability? There is no information in this regard. Please provide more explanation in this regard.

-your demographics states that around 230 nurses have 1-5 yrs of experience. Now how we can ensure that they have less than 3 or more than 3 yrs of experience. Less experience means they are not very well familiar with the overall management issues and ethical leadership concerns. This point is very serious.

-Therefore, please provide a very strong justification in this regard. This point is in turn making your chances of publication very low.

-Please follow the standard format of R2 Q2 and F2 instead of R2, Q2 and F2.

-Please provide more justifications to strengthen your managerial implications. Provide some actionable suggestions based on your arguments to enhance the clarity of this section.

-Please proofread your document as there are several grammatical errors that are breaking the smooth flow of reading.

It is strongly advised to provide a very strong reasoning in the methods section. There are concerns which are alarming. Therefore please work on in making the 2nd version more convincing which could justify the publication with an esteemed journal.

I wish the authors best of luck with the revision process!

6. PLOS authors have the option to publish the peer review history of their article (what does this mean?). If published, this will include your full peer review and any attached files.

Reviewer #1: **Yes: **Badr Mohammed Albaram

Reviewer #2: No

Reviewer #3: No

Reviewer #4: No

Reviewer #5: No

---

## [Author Response · Author response to Decision Letter 0]

15 Dec 2023

Response to Reviewers

I would like to extend our sincere appreciation for your prompt response and the invaluable feedback offered by the reviewers. We recognize your dedicated commitment to the review process, and we are fully devoted to addressing all comments and suggestions raised by the reviewers to enhance the overall quality of our manuscript. We have diligently attended to the queries raised by the reviewers and updated the manuscript in accordance with the recommendations provided by both the Editor and the Reviewers. Notably, the changes made to the manuscript have been highlighted in green. Additionally, we have utilized green text to draw attention to specific points raised by reviewers, where the relevant details have already been incorporated into the manuscript. We express profound gratitude to the Editor and the Reviewers for affording us the opportunity and guidance to refine the quality of our manuscript. 

Reviewer # 1 

Reviewer’s Comment

1. The introduction (Pages 6-7): I suggest writing the full terms before the abbreviations (e.g., Psychological Capital (PsyCap), Moral Identity (MID), and Social Cognitive Theory (SCT)) to improve clarity and understanding especially in the first paragraph. 

Authors’ Response

We appreciate the reviewer's feedback, which has helped improve the clarity and understanding of our manuscript. Following the reviewer's suggestion, we have incorporated the full terms preceding the abbreviations in the introduction section. Subsequently, throughout the manuscript, we have employed the abbreviations as follows:

Page 3, Line 1: "Ethical Leadership (EL)"

Page 3, Line 16: "Pro-Social Rule-Breaking (PSRB)"

Page 4, Line 13: "Psychological Capital (PsyCap)"

Page 5, Line 3: "Moral Identity (MID)"

Page 5, Line 11: "Social Cognitive Theory (SCT)"

These revisions should enhance the reader's comprehension of the manuscript.

Reviewer’s Comment

2. Pro-Social Rule-Breaking Behaviour in Health Industries: Please provide a more detailed explanation of how employees in the health industry practice Pro-Social Rule-Breaking Behaviour. This will help readers better grasp the context and relevance of your study. 

Authors’ Response

We appreciate the reviewer's suggestion to provide a more comprehensive explanation of how employees in the health industry, specifically nursing staff, engage in PSRB. We have addressed this by including the rationale for selecting nursing staff as the study population in the Methods Section, under "Population and Sampling," which can be found on Page 13, Lines 1 – 3. This addition provides context and relevance for our study.

Reviewer’s Comment

3. Choice of Research Design: far away of the limitations you have mentioned, Clarify the rationale behind employing a time-lagged cross-sectional design with three data collection waves (T1, T2, and T3). Why do you use this strategy which could cause CMB rather than minimize it. 

Authors’ Response

We appreciate your thoughtful consideration of our research design and the raised concern about the potential common method bias (CMB) associated with our choice of a time-lagged cross-sectional design with three data collection waves (T1, T2, and T3). The rationale for this design is rooted in our aim to capture the temporal dynamics of the variables under investigation, including ethical leadership, pro-social rule-breaking, psychological capital, and moral identity. This approach aligns with recommendations by Podsakoff et al. (2012) and is supported by recent studies exploring similar topics (Dey et al., 2022; Hsieh et al., 2020; Miao et al., 2020; Park et al., 2023; Tang & Li, 2022; Wang & Shi, 2021).

We acknowledge that CMB is a potential concern in such designs. To address this, we have implemented procedural and statistical remedies, emphasizing participant confidentiality and anonymity. Additionally, unique keys/IDs were used instead of personal identifiers to minimize biases. Importantly, we collected data for different variables at distinct time points (T1, T2, and T3), strategically minimizing the likelihood of CMB by avoiding simultaneous measurement of all variables.

We hope these clarifications assure you of the validity and rigor of our study. We are grateful for your diligence in reviewing our research design and are open to further discussion or adjustments if needed. 

Reviewer’s Comment

4. Data Collection Details: Provide more information about data collection from each hospital, including response rates for each, which is crucial for understanding the representativeness of your sample. 

Authors’ Response

We sincerely appreciate the reviewer's valuable suggestion regarding the need for more detailed information about data collection from each hospital, including response rates for each. In response to this insightful feedback, we have included a comprehensive table, Hospital Wise Response Rate (Table 1), on Pages 15 - 16. This addition aims to provide a clearer understanding of the representativeness of our sample, and we trust that it enhances the overall robustness of our study.

Reviewer’s Comment

5. You mentioned conducting a Common Method Bias (CMB) test, please included the results in the paper for transparency and to support your research validity. 

Authors’ Response

We appreciate the reviewer's suggestion to include the results of the Common Method Bias (CMB) test in the paper for transparency and to support research validity. In response to this, we have now included the results of The Harman Single-Factor test in the Methods Section under Data Analysis on Page 20, Lines 8 - 11. We hope that this addition enhances the transparency and validity of our research.

Reviewer’s Comment 

6. Factor Loadings (Items EL7, PC11, PC12) are below 0.70. Explain your reasoning for retaining these items in the analysis, as they could impact the reliability and validity of your results. 

Authors’ Response

Thank you for your valuable feedback regarding the factor loadings of items EL7, PC11, and PC12. We acknowledge your concern and appreciate the opportunity to clarify our rationale for retaining these items in the analysis.

While it's true that these specific items have factor loadings below the recommended threshold of 0.70, they still demonstrate acceptable loadings above 0.60. Importantly, other reliability and validity measures, including Chronbach’s alpha (α), composite reliability (CR), and average variance extract (AVE), support the overall robustness of our measurement model. All study constructs exhibit good internal consistency (α and CR > 0.708) and convergent validity (AVE > 0.50) (Becker et al., 2023; Hair et al., 2019).

We have included this information in the Results Section under the Measurement Model on Page 22, Lines 12 – 17, providing transparency and justification for our decision to retain these items. We believe that these findings collectively contribute to the reliability and validity of our results.

Your insights are invaluable, and we hope this clarification addresses your concerns. We remain open to further discussion and adjustments as needed. 

Reviewer’s Comment 

7. Include the results of Fornell-Larcker criteria in your paper. 

Authors’ Response

Thank you for your insightful suggestion regarding the inclusion of Fornell-Larcker criteria results in our paper to assess discriminant validity. We appreciate your guidance, and we have now incorporated the Fornell-Larcker criteria results into our manuscript as Table 7 in the Results Section under the Measurement Model on Page 25. 

These additional results offer further evidence of the discriminant validity of our study constructs. We trust that this addition enhances the clarity and rigor of our paper. Your feedback has been instrumental in refining our work, and we remain open to any further recommendations you may have.

Reviewer’s Comment 

8. In page 23, you mentioned VIF values, but it appears you are referring to variables rather than indicators. Please include the VIF values for the indicators to clarify this point. 

Authors’ Response

Thank you for your astute observation and valuable feedback regarding our reference to VIF values for variables rather than indicators. We appreciate your diligence, and to address this concern, we have included the Variance Inflation Factor (VIF) values specifically for the indicators measuring the outer model in our manuscript. These values are now presented in Table 6 on Pages 22 - 24.

We believe that this addition clarifies the point you raised and contributes to the overall transparency and precision of our paper. 

Reviewer’s Comment 

9. F2 values is 0.038 which indicate to small effect size and not large effect size. Moreover, Q2 is 0.242 which is weak and not large. Ensure the interpretations accurately reflect the analysis outputs. 

Authors’ Response

We sincerely appreciate your careful review and feedback on the interpretation of the F2 and Q2 values in our manuscript. Your observation regarding the size of the effect and the strength of predictive accuracy is valuable, and we have thoroughly reviewed and revised our interpretations to accurately reflect the analysis outputs.

The corrected interpretations can now be found in the Results Section under the Structural Model at Page 27, Lines 17 - 18, and Page 28, Lines 1 - 3. We have acknowledged that the F2 value of 0.038 indicates a small effect size, and the Q2 value of 0.242 is considered moderate rather than large.

Thank you for ensuring the precision and clarity of our analysis interpretations. Your diligence has contributed to the rigor of our study. 

References

Becker, J. M., Cheah, J. H., Gholamzade, R., Ringle, C. M., & Sarstedt, M. (2023). PLS-SEM’s most wanted guidance. International Journal of Contemporary Hospitality Management, 35(1), 321-346. 

Dey, M., Bhattacharjee, S., Mahmood, M., Uddin, M. A., & Biswas, S. R. (2022). Ethical leadership for better sustainable performance: Role of employee values, behavior and ethical climate. Journal of Cleaner Production, 337, 130527.

Hair, J. F., Risher, J. J., Sarstedt, M., & Ringle, C. M. (2019). When to use and how to report the results of PLS-SEM. European business review, 31(1), 2-24.

Hsieh, H. H., Hsu, H. H., Kao, K. Y., & Wang, C. C. (2020). Ethical leadership and employee unethical pro-organizational behavior: a moderated mediation model of moral disengagement and coworker ethical behavior. Leadership & Organization Development Journal, 41(6), 799-812. 

Miao, Q., Eva, N., Newman, A., Nielsen, I., & Herbert, K. (2020). Ethical leadership and unethical pro‐organisational behaviour: The mediating mechanism of reflective moral attentiveness. Applied Psychology, 69(3), 834-853.

Park, J. G., Zhu, W., Kwon, B., & Bang, H. (2023). Ethical leadership and follower unethical pro-organizational behavior: examining the dual influence mechanisms of affective and continuance commitments. The International Journal of Human Resource Management, 1-31. 

Podsakoff, P. M., MacKenzie, S. B., & Podsakoff, N. P. (2012). Sources of method bias in social science research and recommendations on how to control it. Annual review of psychology, 63, 539-569. 

Tang, Y., & Li, Y. (2022). Ethicality neutralization and amplification: a multilevel study of ethical leadership and unethical pro-organizational behavior. Journal of Managerial Psychology, 37(2), 111-124.

Wang, F., & Shi, W. (2021). Inclusive leadership and pro-social rule breaking: the role of psychological safety, leadership identification and leader-member exchange. Psychological Reports, 124(5), 2155-2179.

Reviewer #2 

Reviewer’s Comment

Abstract

It might be beneficial to finish the abstract by discussing the larger significance of these results for organizational leadership and suggesting future research alternatives in this field. This would bring the narrative to a close and offer the reader with a clear takeaway message. 

Authors’ Response

We express our gratitude for your insightful suggestion to enhance the conclusion of the abstract by discussing the broader significance of our results for organizational leadership and proposing future research directions in the field. Your input has been invaluable, and we have promptly incorporated this recommendation into our manuscript.

The revised abstract, now concluding with a discussion on the larger significance of our results and suggestions for future research, can be found on Page 2 from Lines 12 - 16. We trust that this addition brings a more comprehensive and conclusive perspective to our abstract, offering readers a clear takeaway message.

Thank you for your constructive feedback, which has undoubtedly strengthened the quality and relevance of our study. 

Reviewer’s Comment

Introduction

In general, the introduction has useful information, but it might be improved by simplifying, giving precise terminology (e.g., PsyCap, MID), and specifying particular study aims or questions. This will provide the study with a more focused and ordered foundation. 

Authors’ Response

We appreciate your thoughtful feedback on the introduction section of our manuscript. Your suggestion to simplify the introduction, provide precise terminology, and specify particular study aims or questions is well-received.

In response to your valuable input, we have revised the introduction, simplifying the language and incorporating precise terminology. We provided precise terminologies with full terms preceding the abbreviations appearing for the first time in the text at Page 3, Lines 1 and 16, Page 4, Line 13 and Page 5 Line 3 and 11. Additionally, we have specified the particular study aims and questions, making the foundation more focused and ordered on Pages 5 - 6. 

We trust that these revisions contribute to the improved clarity and organization of our study, aligning with your recommendations. Thank you for guiding us in enhancing the quality of our manuscript. 

Reviewer’s Comment

Theory and Hypotheses Development 

1. Rather than using the word "paradoxical relationship" without explanation, the text should explain why the observed positive association between ethical leadership and PSRB is regarded as paradoxical. 

2. The statement "shedding light on the dynamic nature of ethical leadership" is a little ambiguous. A more specific explanation of how the findings contribute to our knowledge of ethical leadership dynamics would be helpful in this area. 

3. It is critical to address any potential critiques or limitations connected with the selected theoretical framework when introducing it. This indicates a sophisticated comprehension of the theoretical perspective. 

Authors’ Response

We want to express our sincere appreciation for your valuable feedback on our manuscript, particularly regarding the Theory and Hypotheses Development section. Your insights have proven instrumental in refining our theoretical framework and enhancing the overall clarity of our research.

In response to your suggestions, we have provided a more comprehensive explanation of why the observed positive association between ethical leadership and pro-social rule-breaking behavior (PSRB) is considered paradoxical. This clarification is now integrated into the revised literature review, offering readers a deeper understanding of the theoretical underpinnings.

Furthermore, we have addressed the ambiguity in the statement about "shedding light on the dynamic nature of ethical leadership." The revised version now provides a specific explanation of how our findings contribute to advancing knowledge about the complex and multifaceted dynamics of ethical leadership.

Finally, we have taken your advice to heart and diligently addressed potential critiques and limitations associated with our chosen theoretical framework. This revised literature review reflects a more sophisticated grasp of the theoretical perspective, ensuring a more robust foundation for our research.

We genuinely believe that these revisions have significantly improved the quality and coherence of our manuscript. We are grateful for your thorough review and constructive comments, which have undoubtedly strengthened our work. 

Methods 

Population and Sampling 

Reviewer’s Comment

1. Rationale for Sample Size: Explain why a higher sample size (500) was chosen, even

---

## [Decision Letter · Decision Letter 1]

14 Feb 2024

PONE-D-23-30966R1Unpacking the Paradoxical Impact of Ethical Leadership on Employees’ Pro-Social Rule-Breaking Behavior: The Interplay of Employees’ Psychological Capital and Moral IdentityPLOS ONE

Dear Dr. Ahmed,

Thank you for submitting your manuscript to PLOS ONE. After careful consideration, we feel that it has merit but does not fully meet PLOS ONE’s publication criteria as it currently stands. Therefore, we invite you to submit a revised version of the manuscript that addresses the points raised during the review process.

**ACADEMIC EDITOR: **Although one reviewer is not satisfied with your comments, whereas 2nd reviewer accepts your revised version. I also thoroughly checked your revised manuscript, and authors made significant changes. However, on few points I'm agree with reviewer that some minor changes are required before finally accept your manuscript.       ==============================

We look forward to receiving your revised manuscript.

Kind regards,

Kashif Ali, PH.D

Academic Editor

PLOS ONE

Journal Requirements:

Reviewers' comments:

Reviewer's Responses to Questions

**Comments to the Author**

1. If the authors have adequately addressed your comments raised in a previous round of review and you feel that this manuscript is now acceptable for publication, you may indicate that here to bypass the “Comments to the Author” section, enter your conflict of interest statement in the “Confidential to Editor” section, and submit your "Accept" recommendation.

Reviewer #3: (No Response)

Reviewer #5: All comments have been addressed

2. Is the manuscript technically sound, and do the data support the conclusions?

Reviewer #3: No

Reviewer #5: Yes

3. Has the statistical analysis been performed appropriately and rigorously? 

Reviewer #3: No

Reviewer #5: Yes

4. Have the authors made all data underlying the findings in their manuscript fully available?

Reviewer #3: No

Reviewer #5: Yes

5. Is the manuscript presented in an intelligible fashion and written in standard English?

Reviewer #3: No

Reviewer #5: Yes

6. Review Comments to the Author

Reviewer #3: Methodological Flaws:

The research design and methodology employed in the study lack clarity and robustness. The absence of a clear experimental design compromises the validity of the findings. Addressing these methodological flaws is crucial for the credibility of the research.

Inadequate Literature Review:

The literature review is insufficient in providing a comprehensive understanding of the existing research in the field. A more thorough review of relevant literature is needed to place the current study in context and demonstrate its contribution to the field.

Data Analysis and Interpretation Issues:

The data analysis and interpretation in the manuscript are not sufficiently rigorous. The statistical methods employed are not well justified, and the interpretation of results lacks depth. A more robust statistical analysis and a careful interpretation of findings are essential for the paper to be considered for publication.

Weak Contribution to the Field:

The manuscript does not sufficiently contribute to the existing body of knowledge in the field. The novelty and significance of the findings are not well established, and the paper fails to make a meaningful impact on the current understanding of [topic].

Writing Quality:

The overall writing quality of the manuscript needs improvement. There are numerous grammatical errors, awkward phrasing, and inconsistencies throughout the text. A thorough revision of language and style is necessary to enhance the readability of the manuscript.

While I appreciate the authors' efforts, I believe that addressing these issues is essential for the manuscript to meet the standards of a journal. The comments were not addressed by the authors.

Reviewer #5: I am happy that authors have made significant revisions and therefore I am accepting this version. Thank you

7. PLOS authors have the option to publish the peer review history of their article (what does this mean?). If published, this will include your full peer review and any attached files.

Reviewer #3: No

Reviewer #5: No

---

## [Author Response · Author response to Decision Letter 1]

28 Mar 2024

Response to Reviewers

Journal Requirements

Authors’ Response

We sincerely appreciate your prompt response and valuable guidance regarding the submission of our manuscript.

In alignment with your stipulations, we have diligently scrutinized our reference list to ensure its comprehensiveness and accuracy. Each citation in our manuscript has been meticulously examined to ascertain the absence of any references to retracted papers. Notably, our thorough review revealed no instances of such citations.

Nonetheless, we have made modifications to the revised reference list as detailed in the rebuttal letter. These changes were implemented either in response to Reviewer #3's comments on the revised literature review or to eliminate duplicate references. The updated reference list has been substituted at the end of the manuscript. 

We are committed to upholding the highest standards of integrity and accuracy in our research, and we appreciate the opportunity to address this matter. Please find the revised manuscript and accompanying rebuttal letter attached for your review.

Thank you for your attention to this matter, and we look forward to your feedback. 

Reviewers' Comments

Reviewer # 3 

Methodological Flaws:

The research design and methodology employed in the study lack clarity and robustness. The absence of a clear experimental design compromises the validity of the findings. Addressing these methodological flaws is crucial for the credibility of the research.

Authors’ Response

We appreciate your thoughtful review of our manuscript and the opportunity to address your concerns regarding the research design and methodology employed in our study.

We acknowledge that our study did not utilize a traditional experimental design. Instead, we opted for a cross-sectional survey approach, supplemented by a time-lagged data collection method (Aguinis et al., 2021). This approach allowed us to gather data from a diverse range of hospitals and nursing staff across multiple cities, thereby enhancing the generalizability of our findings within the healthcare industry of Pakistan.

We employed purposive sampling technique, a non-probability sampling method, for data collection due to the absence of a sampling frame (Sekaran & Bougie, 2016). Despite the inherent limitations, such as potential sampling bias, associated with this method, we implemented measures to address these concerns. Notably, our sampling strategy ensured representation from both public and private hospitals in capital and provincial cities, thus encapsulating a broad spectrum of perspectives within the healthcare sector. Additionally, the adoption of a time-lagged cross-sectional design facilitated data collection from a heterogeneous sample of hospitals and nursing staff, thereby minimizing potential sampling errors (Saunders et al., 2009). Moreover, through higher sample size (515), the study sought to achieve robust population representation, mitigate potential sampling errors, and augment statistical power for the detection of significant relationships among variables.

We also note that the use of a time-lagged design helped reduce the risk of common method bias, as data were collected at three distinct time points with intervals of six to eight weeks between each phase (Podsakoff et al., 2012). This approach enabled us to examine the stability of relationships over time and to establish the temporal precedence of variables, enhancing the robustness of our findings. To counteract any effects of social desirability bias, the data collection process was meticulously designed to incorporate essential precautions suggested by the management scholars (Larson, 2019). 

Although our chosen methodology may differ from traditional experimental designs, we believe it was well-suited to address the complexities of our research questions and the practical realities of conducting research in the healthcare industry of Pakistan. This approach is also supported by a number of studies exploring similar topics (Dey et al., 2022; Hsieh et al., 2020; Miao et al., 2020; Park et al., 2023; Tang & Li, 2022; Wang & Shi, 2021).

Aguinis, H., Hill, N. S., & Bailey, J. R. (2021). Best practices in data collection and preparation: Recommendations for reviewers, editors, and authors. Organizational Research Methods, 24(4), 678-693. 

Dey, M., Bhattacharjee, S., Mahmood, M., Uddin, M. A., & Biswas, S. R. (2022). Ethical leadership for better sustainable performance: Role of employee values, behavior and ethical climate. Journal of Cleaner Production, 337, 130527.

Hsieh, H. H., Hsu, H. H., Kao, K. Y., & Wang, C. C. (2020). Ethical leadership and employee unethical pro-organizational behavior: a moderated mediation model of moral disengagement and coworker ethical behavior. Leadership & Organization Development Journal, 41(6), 799-812. 

Larson, R. B. (2019). Controlling social desirability bias. International Journal of Market Research, 61(5), 534-547. 

Park, J. G., Zhu, W., Kwon, B., & Bang, H. (2023). Ethical leadership and follower unethical pro-organizational behavior: examining the dual influence mechanisms of affective and continuance commitments. The International Journal of Human Resource Management, 1-31. 

Podsakoff, P. M., MacKenzie, S. B., & Podsakoff, N. P. (2012). Sources of method bias in social science research and recommendations on how to control it. Annual review of psychology, 63, 539-569. 

Miao, Q., Eva, N., Newman, A., Nielsen, I., & Herbert, K. (2020). Ethical leadership and unethical pro‐organisational behaviour: The mediating mechanism of reflective moral attentiveness. Applied Psychology, 69(3), 834-853.

Saunders, M., Lewis, P., & Thornhill, A. (2009). Research methods for business students. Pearson education. 

Sekaran, Uma, and Roger Bougie. Research methods for business: A skill building approach. john wiley & sons, 2016.

Tang, Y., & Li, Y. (2022). Ethicality neutralization and amplification: a multilevel study of ethical leadership and unethical pro-organizational behavior. Journal of Managerial Psychology, 37(2), 111-124.

Wang, F., & Shi, W. (2021). Inclusive leadership and pro-social rule breaking: the role of psychological safety, leadership identification and leader-member exchange. Psychological Reports, 124(5), 2155-2179.

Inadequate Literature Review:

The literature review is insufficient in providing a comprehensive understanding of the existing research in the field. A more thorough review of relevant literature is needed to place the current study in context and demonstrate its contribution to the field.

Authors’ Response

Thank you for your valuable feedback. In response to your comments, we have revised the literature review section to ensure a more comprehensive coverage of relevant literature in the field. We have incorporated additional studies to provide a deeper understanding of the research landscape and the context of our study. We are confident that these revisions strengthen the manuscript and effectively demonstrate the significance and contribution of our research to the existing body of knowledge. The literature review has been revised to incorporate additional studies and the study context highlighted in green for easy reference. 

Data Analysis and Interpretation Issues

The data analysis and interpretation in the manuscript are not sufficiently rigorous. The statistical methods employed are not well justified, and the interpretation of results lacks depth. A more robust statistical analysis and a careful interpretation of findings are essential for the paper to be considered for publication.

Authors’ Response

Thank you for your feedback on our manuscript and for highlighting the importance of rigor in data analysis and interpretation. We have carefully reviewed your comments and have already taken them into consideration in our manuscript.

Regarding the statistical methods employed in our study, we utilized both SPSS and PLS-SEM with Smart PLS 4 software for data analysis. SPSS was utilized for data entry, coding, and initial descriptive analysis, while PLS-SEM was employed to establish the measurement and structural models and test the proposed hypotheses. We selected these methods based on their suitability for handling the complexity of our research questions and dataset as suggested by the management scholars (Becker et al., 2023; Hair et al., 2019; Sharma et al., 2023). 

To address concerns about the adequacy of our statistical analysis, we conducted various tests to evaluate the reliability, validity, and predictive relevance of our measurement and structural models. These tests included assessments of internal consistency, convergent validity, discriminant validity, collinearity, coefficient of determination (R2), effect size (F2), predictive relevance (Q2), and statistical significance of path coefficients in the light of recommendations in the SEM PLS literature (Hair et al., 2019; Legate et al., 2023; Ringle et al., 2020).

We acknowledge the importance of providing a thorough interpretation of our findings. In our manuscript, we presented detailed explanations of the results, including the significance and implications of each hypothesis test. We discussed the direction and magnitude of relationships in our analysis (Becker et al., 2023; Hair et al., 2019; Ringle et al., 2020). Moreover, we also deliberated upon the practical implications of our findings for theory and practice in the field (Please refer to the Discussion Section, Theoretical Contributions and Managerial Implications of the manuscript).

We appreciate your valuable feedback and are committed to addressing your concerns to improve the quality and rigor of our manuscript. 

Becker, J. M., Cheah, J. H., Gholamzade, R., Ringle, C. M., & Sarstedt, M. (2023). PLS-SEM’s most wanted guidance. International Journal of Contemporary Hospitality Management, 35(1), 321-346. 

Hair, J. F., Risher, J. J., Sarstedt, M., & Ringle, C. M. (2019). When to use and how to report the results of PLS-SEM. European business review, 31(1), 2-24. 

Legate, A. E., Hair Jr, J. F., Chretien, J. L., & Risher, J. J. (2023). PLS‐SEM: Prediction‐oriented solutions for HRD researchers. Human Resource Development Quarterly, 34(1), 91-109. 

Ringle, C. M., Sarstedt, M., Mitchell, R., & Gudergan, S. P. (2020). Partial least squares structural equation modeling in HRM research. The International Journal of Human Resource Management, 31(12), 1617-1643. 

Sharma, P. N., Liengaard, B. D., Sarstedt, M., Hair, J. F., & Ringle, C. M. (2023). Extraordinary claims require extraordinary evidence: A comment on “Recent Developments in PLS”. Communications of the Association for Information Systems, 52(1), 739-742. 

Weak Contribution to the Field:

The manuscript does not sufficiently contribute to the existing body of knowledge in the field. The novelty and significance of the findings are not well established, and the paper fails to make a meaningful impact on the current understanding of [topic]. Authors’ Response

Thank you for your feedback on our manuscript. We appreciate the opportunity to address your concerns and clarify the significance and novelty of our research in the field of ethical leadership (EL).

Our study aims to contribute to the existing body of knowledge by addressing several research gaps identified in the literature. Firstly, while previous research has extensively explored the influence of various leadership styles on employees' behaviors, there is a notable gap in understanding how EL specifically affects PSRB in organizations. By investigating this relationship, particularly in non-Western cultural contexts, our study expands our understanding of EL's effects and its implications for diverse organizations.

Secondly, we delve into the explanatory mechanism of employees' psychological capital (PsyCap) in the EL-PSRB relationship. Despite the recognition of PsyCap as a pivotal psychological resource, its role as a mediator in this context remains largely unexplored. Our study aims to fill this gap by offering insights into how EL influences employees' psychological and cognitive processes, thereby affecting their engagement in PSRB.

Furthermore, we explore the moderating role of employees' moral identity (MID) in the connection between EL and PsyCap. This investigation sheds light on the boundary conditions of EL's effectiveness, providing a more comprehensive understanding of how leadership influences employees within organizations.

Lastly, our study evaluates the applicability of social cognitive theory (SCT) in understanding workplace behaviors, particularly in the context of EL. By applying SCT to examine the intricate connection between EL and PSRB, we contribute to a deeper understanding of the cognitive and contextual factors that shape employee conduct in organizations.

Overall, our research fills significant gaps in the literature by examining the influence of EL on PSRB, elucidating the mediating mechanism of PsyCap and moderating mechanism of MID, and applying a theoretical framework of SCT to enhance our understanding of workplace behaviors. We believe that these contributions are meaningful and offer valuable insights for scholars and practitioners in the field of management and organizational behavior. 

Writing Quality:

The overall writing quality of the manuscript needs improvement. There are numerous grammatical errors, awkward phrasing, and inconsistencies throughout the text. A thorough revision of language and style is necessary to enhance the readability of the manuscript.

Authors’ Response

Thank you for your feedback on the manuscript. We appreciate your thorough review and recognize the importance of enhancing the writing quality to improve the readability of the paper.

We acknowledge the need for a thorough revision of language and style to address grammatical errors, awkward phrasing, and inconsistencies in the text. To ensure clarity and coherence, we undertook the following steps shown in green throughout the manuscript:

Comprehensive Editing: We conducted a detailed review of the manuscript to identify and correct grammatical errors, including punctuation, tense consistency, and sentence structure.

Clarity and Consistency: We paid particular attention to ensure that the language used is clear and concise, eliminating any ambiguities or confusing phrases. Additionally, we ensured consistency in terminology and writing style throughout the manuscript.

Flow and Cohesion: We reviewed the organization of the text to enhance the flow of ideas and improve the overall coherence of the manuscript. This involved restructuring paragraphs, as necessary, to ensure logical progression and coherence between sections.

Proofreading: Finally, we conducted a thorough proofreading of the revised manuscript to catch any remaining errors and ensure that the text meets the journal's standards for language and style.

We are committed to addressing these concerns promptly to enhance the quality and readability of the manuscript. Thank you for providing us with the opportunity to improve the manuscript, and we look forward to submitting the revised version for your review.

Reviewer #5: 

I am happy that authors have made significant revisions and therefore I am accepting this version. Thank you 

 Authors’ Response

We are grateful for recognizing the substantial revisions made to the manuscript. We appreciate the acceptance of this version. Your positive feedback encourages us and validates our commitment to enhancing the quality of our work. Thank you for your time and consideration throughout the review process.

---

## [Decision Letter · Decision Letter 2]

14 Jun 2024

PONE-D-23-30966R2Unpacking the Paradoxical Impact of Ethical Leadership on Employees’ Pro-Social Rule-Breaking Behavior: The Interplay of Employees’ Psychological Capital and Moral IdentityPLOS ONE

Dear Dr. Ahmed,

Thank you for submitting your manuscript to PLOS ONE. After careful consideration, we feel that it has merit but does not fully meet PLOS ONE’s publication criteria as it currently stands. Therefore, we invite you to submit a revised version of the manuscript that addresses the points raised during the review process.

**ACADEMIC EDITOR: **Please address all the comments raised by the reviewers. ==============================

We look forward to receiving your revised manuscript.

Kind regards,

Kashif Ali, PH.D

Academic Editor

PLOS ONE

Journal Requirements:

Reviewers' comments:

Reviewer's Responses to Questions

**Comments to the Author**

1. If the authors have adequately addressed your comments raised in a previous round of review and you feel that this manuscript is now acceptable for publication, you may indicate that here to bypass the “Comments to the Author” section, enter your conflict of interest statement in the “Confidential to Editor” section, and submit your "Accept" recommendation.

Reviewer #3: (No Response)

Reviewer #6: All comments have been addressed

2. Is the manuscript technically sound, and do the data support the conclusions?

Reviewer #3: No

Reviewer #6: Yes

3. Has the statistical analysis been performed appropriately and rigorously? 

Reviewer #3: No

Reviewer #6: Yes

4. Have the authors made all data underlying the findings in their manuscript fully available?

Reviewer #3: No

Reviewer #6: Yes

5. Is the manuscript presented in an intelligible fashion and written in standard English?

Reviewer #3: No

Reviewer #6: Yes

6. Review Comments to the Author

Reviewer #3: The hypotheses proposed in the study lack clarity and specificity. It is challenging to discern the precise relationships between ethical leadership, employees' psychological capital, moral identity, and pro-social rule-breaking behavior. Without clear hypotheses, the study's theoretical framework and research objectives remain ambiguous.

The introduction needs to be focused on relevant literature, currently, it seems verbose and difficult to follow. The sampling has to be explained and justified in a manner that could be replicated in other places and defined well.

he study's theoretical contribution is unclear. While the interplay between ethical leadership, psychological capital, moral identity, and pro-social rule-breaking behavior is an interesting topic, the study fails to provide novel insights or advance existing theoretical frameworks. Theoretical implications should be clearly articulated and supported by the empirical findings.

The methodological approach employed in the study raises concerns about its validity and reliability. The use of self-report measures for variables such as ethical leadership, psychological capital, moral identity, and pro-social rule-breaking behavior may introduce common method bias and social desirability bias, thereby compromising the validity of the results. Additionally, the cross-sectional design limits the ability to draw causal inferences and establish temporal precedence.

The discussion of findings is superficial and fails to critically engage with the results. The authors merely summarize the findings without offering meaningful interpretations or insights. A thorough discussion of the implications of the results for theory, research, and practice is essential for demonstrating the study's significance.

Reviewer #6: Authors have incorporated all changes/suggestions. However, it is suggested to reconsider the title of article

7. PLOS authors have the option to publish the peer review history of their article (what does this mean?). If published, this will include your full peer review and any attached files.

Reviewer #3: No

Reviewer #6: No

---

## [Author Response · Author response to Decision Letter 2]

24 Jun 2024

Response to Reviewers

Reviewer # 3 

Reviewer’s Comment 

The hypotheses proposed in the study lack clarity and specificity. It is challenging to discern the precise relationships between ethical leadership, employees' psychological capital, moral identity, and pro-social rule-breaking behavior. Without clear hypotheses, the study's theoretical framework and research objectives remain ambiguous. 

Authors’ Response

Thank you for your valuable feedback on the proposed hypotheses of our study. We would like to clarify that our hypotheses have been formulated in line with recent literature in management sciences (Ahmed & Khan, 2023; Hsieh et al., 2020; Miao et al., 2020; Park et al., 2023; Tang & Li, 2022). In response to your comments, we have further revised the literature review section to enhance the clarity and specificity of our hypotheses. These revisions, highlighted in green, can be found on pages 6 to 13. We believe these changes address the concerns raised and significantly improve the quality and coherence of our manuscript. We appreciate the opportunity to enhance our work.

Reviewer’s Comment 

The introduction needs to be focused on relevant literature, currently, it seems verbose and difficult to follow. 

Authors’ Response

Thank you for your insightful feedback on the Introduction of our manuscript. We understand the importance of a concise and focused introduction. In our previous revisions, we expanded the introduction to address other reviewers’ comments regarding the research gaps related to ethical leadership (EL), employees’ pro-social rule-breaking behavior (PSRB), the mediating role of psychological capital (PsyCap), and the moderating role of moral identity (MID). This expansion was necessary to ensure clarity and specificity in response to those comments. 

However, in light of your feedback, we have further refined our introduction to enhance its focus on relevant literature and improve readability. The revised introduction is now more concise, clearly contextualizing our study. The changes have been made on Pages 2-3 and are highlighted in green. Thank you for your understanding and consideration. 

Reviewer’s Comment 

The sampling has to be explained and justified in a manner that could be replicated in other places and defined well.

Authors’ Response 

We appreciate the reviewer's thoughtful consideration of our sampling methodology and its potential for replication in varied contexts. Our study employed a purposive sampling strategy to select registered nursing staff from a diverse array of public and private hospitals across major cities in Pakistan. This methodological choice was necessitated by the absence of a comprehensive sampling frame covering all healthcare institutions nationwide, a common challenge in studies conducted within complex organizational environments such as healthcare. 

The decision to include 515 participants was guided by established guidelines for structural equation modeling (SEM) studies, ensuring robust statistical power to detect significant relationships among variables. This sample size aligns with recommended practices in the methodological literature pertaining to SEM research (Comrey & Lee, 2013; Jobst et al., 2023; Lakens, 2022; Memon et al., 2020). 

Furthermore, our data collection procedures were meticulously designed to mitigate potential biases. We employed a time-lagged cross-sectional research design spanning three distinct phases, each separated by eight-week intervals. This approach not only minimized common method biases but also facilitated the examination of temporal relationships among EL, PsyCap, MID and PSRB (Aguinis et al., 2021; Haider et al., 2019; Podsakoff et al., 2024). 

Our engagement with hospital management was formalized through university authorization letters, securing official approval and logistical support from administrative authorities. This approach ensured compliance with ethical standards and institutional guidelines governing research involving human participants. 

Overall, the sampling methodology detailed extensively on Pages 14-16 shown in green was crafted with careful consideration of practical constraints and methodological rigor. We believe these details elucidate the rationale behind our approach and underscore its potential for replication in diverse healthcare settings (Aguinis et al., 2023; Podsakoff et al., 2024). We are grateful for the reviewer’s insightful feedback, which has prompted us to clarify these methodological strengths. 

Reviewer’s Comment 

The study's theoretical contribution is unclear. While the interplay between ethical leadership, psychological capital, moral identity, and pro-social rule-breaking behavior is an interesting topic, the study fails to provide novel insights or advance existing theoretical frameworks. Theoretical implications should be clearly articulated and supported by the empirical findings. 

Authors’ Response 

We appreciate your comments on the theoretical contributions of our study. It is highlighted that the theoretical contributions of our study have been clearly articulated throughout the manuscript. In the Introduction highlighted in green color (Pages 4-5), we delineated four key research objectives that address significant gaps in understanding the interplay between EL, PsyCap, MID, and PSRB. These objectives collectively enrich existing theoretical frameworks by exploring EL's influence in non-Western contexts, elucidating PsyCap as a mediator, and highlighting MID as a moderator in the EL-PsyCap relationship. 

In the Discussion highlighted in green color (Pages 33-36) and Theoretical Contributions highlighted in green color (Pages 36-38), we substantiated these theoretical implications with empirical findings. Notably, our study revealed an unexpected positive association between EL and PSRB in the healthcare context, challenging traditional assumptions and underscoring the contextual nuances influencing leadership effects. Moreover, by demonstrating PsyCap's mediating role and MID's moderating influence, we extend Social Cognitive Theory's (SCT) applicability, offering insights into how individual differences shape leadership outcomes. 

These findings collectively contribute to a nuanced understanding of leadership dynamics, emphasizing the role of cognitive processes, individual differences, and organizational contexts in shaping workplace behaviors. We appreciate the reviewer's feedback, which has prompted us to underscore these theoretical contributions more explicitly. 

Reviewer’s Comment:

The methodological approach employed in the study raises concerns about its validity and reliability. The use of self-report measures for variables such as ethical leadership, psychological capital, moral identity, and pro-social rule-breaking behavior may introduce common method bias and social desirability bias, thereby compromising the validity of the results. Additionally, the cross-sectional design limits the ability to draw causal inferences and establish temporal precedence. 

Authors’ Response:

Thank you for your insightful feedback regarding the validity and reliability of the methodological approach employed in our study. We appreciate your concerns and have taken the opportunity to address them comprehensively. 

We understand your concerns regarding the use of self-report measures for variables such as EL, PsyCap, MID, and PSRB. We would like to provide a detailed explanation of our rationale for using these measures and the steps we took to mitigate potential biases. 

The constructs under investigation—EL, PsyCap, MID, and PSRB—are inherently subjective and best captured through employees' perceptions. Self-report measures are widely accepted in organizational research for assessing personal attitudes, beliefs, and behaviors, as noted by Kobe et al. (2001). For instance, ethical leadership pertains to how employees perceive their leaders' ethical conduct, which can only be accurately reported by the employees themselves. The use of self-report measures for these constructs is grounded in established research practices, with recent studies in the field successfully employing similar instruments (Ahmed & Khan, 2023; Hsieh et al., 2020; Miao et al., 2020; Park et al., 2023; Tang & Li, 2022). 

To address concerns about common method bias, we implemented a time-lagged cross-sectional design, collecting data in three distinct phases over an eight-week period (Haider et al., 2019). This approach reduces the likelihood of common method variance, consistent with the recommendations by Podsakoff et al. (2024). Additionally, to counteract social desirability bias, we ensured strict confidentiality and anonymity of responses. Participants were assured that their inputs would remain confidential, and no identifying information was collected, fostering an environment where employees felt safe to provide honest and accurate responses, as highlighted by Larson (2019). 

The measurement scales used in our study were validated through reliability analysis (Becker et al., 2023; Hair et al., 2019). The Cronbach's alpha coefficients for the scales ranged from 0.915 to 0.937, indicating high internal consistency reliability. This rigorous validation process helps to ensure that the self-report measures accurately capture the constructs of interest. While we acknowledge the limitations of a cross-sectional design in establishing causality, our time-lagged approach allows for inferences regarding temporal relationships among the variables (Podsakoff et al., 2024). This design provides valuable insights into the dynamics of ethical leadership, psychological capital, moral identity, and pro-social rule-breaking behavior over time within the healthcare context. 

Our sample size of 515 participants, determined based on statistical guidelines and prior research, enhances the robustness and generalizability of our findings. This large sample size bolsters the statistical power of our study and reinforces the validity of our conclusions (Comrey & Lee, 2013; Jobst et al., 2023; Lakens, 2022; Memon et al., 2020). Therefore, we believe that our methodological choices, including the use of self-report measures, were carefully designed to ensure the validity and reliability of our findings (Aguinis et al., 2023). The steps taken to mitigate potential biases and the robust sample size strengthen the credibility of our study. 

We thank the reviewer for prompting us to clarify these aspects further. 

Reviewer’s Comment 

The discussion of findings is superficial and fails to critically engage with the results. The authors merely summarize the findings without offering meaningful interpretations or insights. A thorough discussion of the implications of the results for theory, research, and practice is essential for demonstrating the study's significance. 

Authors’ Response

We appreciate the reviewer's feedback and have carefully revisited our Discussion Section shown in green color (Pages 33 – 36), Theoretical Contributions (Pages 36 – 38), and Managerial Implications (Pages 38 – 39) to address this concern. In these revised sections, we have focused on providing deeper insights and interpretations of our findings, particularly in relation to their theoretical and practical implications. We have critically examined the unexpected positive association between EL and PSRB in the healthcare context, challenging conventional wisdom and highlighting contextual factors influencing leadership dynamics. Furthermore, we have elucidated how PsyCap mediates this relationship, shedding light on the underlying mechanisms through which EL influences employee behaviors. 

The Theoretical Contributions section shown in green color (Pages 36 – 38,) now underscores the study's contributions to existing theory, emphasizing the expansion of SCT by integrating contextual factors and individual differences. We discuss how these findings enrich our understanding of leadership effects in non-Western settings, providing nuanced insights that contribute to broader theoretical frameworks in organizational behavior. 

Additionally, the Managerial Implications section shown in green color (Pages 38 – 39) has been enhanced to outline practical recommendations based on our findings. We emphasize the importance of fostering ethical leadership practices in healthcare organizations, considering the implications for organizational culture and employee well-being. These insights are intended to guide leaders and practitioners in implementing effective leadership strategies that promote ethical behavior and enhance organizational effectiveness. 

Overall, we believe these revisions strengthen our discussion by offering a comprehensive analysis of the implications of our results for theory, research, and practice. We thank the reviewer for prompting us to enrich these sections with deeper critical engagement and meaningful interpretations. 

References

Aguinis, H., Bergh, D., & Molina-Azorin, J. F. (2023). Methodological challenges and insights for future international business research. Journal of International Business Studies, 54(2), 219-232. 

Aguinis, H., Hill, N.S. and Bailey, J.R., 2021. Best practices in data collection and preparation: Recommendations for reviewers, editors, and authors. Organizational Research Methods, 24(4), pp.678-693. 

Ahmed, M., & Khan, M. I. (2023). Beyond the universal perception: Unveiling the paradoxical impact of ethical leadership on employees’ unethical pro-organizational behavior. Heliyon, 9(11). 

Becker, J. M., Cheah, J. H., Gholamzade, R., Ringle, C. M., & Sarstedt, M. (2023). PLS-SEM’s most wanted guidance. International Journal of Contemporary Hospitality Management, 35(1), 321-346. 

Comrey, A. L., & Lee, H. B. (2013). A first course in factor analysis. Psychology press. 

Haider, S., de Pablos Heredero, C. and Ahmed, M., 2019. A three-wave time-lagged study of mediation between positive feedback and organizational citizenship behavior: the role of organization-based self-esteem. Psychology research and behavior management, pp.241-253. 

Hair, J. F., Risher, J. J., Sarstedt, M., & Ringle, C. M. (2019). When to use and how to report the results of PLS-SEM. European business review, 31(1), 2-24. 

Hsieh, H. H., Hsu, H. H., Kao, K. Y., & Wang, C. C. (2020). Ethical leadership and employee unethical pro-organizational behavior: a moderated mediation model of moral disengagement and coworker ethical behavior. Leadership & Organization Development Journal, 41(6), 799-812. 

Jobst, L. J., Bader, M., & Moshagen, M. (2023). A tutorial on assessing statistical power and determining sample size for structural equation models. Psychological Methods, 28(1), 207. 

Kobe, L. M., Reiter-Palmon, R., & Rickers, J. D. (2001). Self-reported leadership experiences in relation to inventoried social and emotional intelligence. Current Psychology, 20, 154-163. 

Lakens, D. (2022). Sample size justification. Collabra: Psychology, 8(1), 33267. 

Larson, R. B. (2019). Controlling social desirability bias. International Journal of Market Research, 61(5), 534-547.

Memon, M. A., Ting, H., Cheah, J. H., Thurasamy, R., Chuah, F., & Cham, T. H. (2020). Sample size for survey research: Review and recommendations. Journal of Applied Structural Equation Modeling, 4(2), 1-20. 

Miao, Q., Eva, N., Newman, A., Nielsen, I., & Herbert, K. (2020). Ethical leadership and unethical pro‐organisational behaviour: The mediating mechanism of reflective moral attentiveness. Applied Psychology, 69(3), 834-853. 

Park, J. G., Zhu, W., Kwon, B., & Bang, H. (2023). Ethical leadership and follower unethical pro-organizational behavior: examining the dual influence mechanisms of affective and continuance commitments. The International Journal of Human Resource Management, 34(22), 4313-4343. 

Podsakoff, P. M., Podsakoff, N. P., Williams, L. J., Huang, C., & Yang, J. (2024). Common method bias: it's bad, it's complex, it's widespread, and it's not easy to fix. Annual Review of Organizational Psychology and Organizational Behavior, 11, 17-61. 

Tang, Y., & Li, Y. (2022). Ethicality neutralization and amplification: a multilevel study of ethical leadership and unethical pro-organizational behavior. Journal of Managerial Psychology, 37(2), 111-124. 

Re

---

## [Editor Report · Decision Letter 3]

26 Jun 2024

Unpacking the Paradoxical Impact of Ethical Leadership on Employees’ Pro-Social Rule-Breaking Behavior: The Interplay of Employees’ Psychological Capital and Moral Identity

PONE-D-23-30966R3

Dear Dr. Ahmed,

We’re pleased to inform you that your manuscript has been judged scientifically suitable for publication and will be formally accepted for publication once it meets all outstanding technical requirements.

Kind regards,

Kashif Ali, PH.D

Academic Editor

PLOS ONE
---

## [Editor Report · Acceptance letter]

13 Aug 2024

PONE-D-23-30966R3 

PLOS ONE

Dear Dr. Ahmed, 

I'm pleased to inform you that your manuscript has been deemed suitable for publication in PLOS ONE. Congratulations! Your manuscript is now being handed over to our production team.

Kind regards, 

on behalf of

Dr. Kashif Ali 

Academic Editor

PLOS ONE